# Preference-Based Dynamic Ranking Structure Recognition

**Nan Lu**[1,2]
nlu99007@gmail.com

**Jian Shi**[1,2,*]
jshi@iss.ac.cn

**Xin-Yu Tian**[3]
tianx@umn.edu

[1]State Key Laboratory of Mathematical Sciences, Academy of Mathematics and Systems Science, Chinese Academy of Sciences, Beijing, China
[2]School of Mathematical Sciences, University of Chinese Academy of Sciences, Beijing, China
[3]School of Statistics, University of Minnesota, Minneapolis, USA

## Abstract

Preference-based data often appear complex and noisy but may conceal underlying homogeneous structures. This paper introduces a novel framework of ranking structure recognition for preference-based data. We first develop an approach to identify dynamic ranking groups by incorporating temporal penalties into a spectral estimation for the celebrated Bradley-Terry model. To detect structural changes, we introduce an innovative objective function and present a practicable algorithm based on dynamic programming. Theoretically, we establish the consistency of ranking group recognition by exploiting properties of a random 'design matrix' induced by a reversible Markov chain. We also tailor a group inverse technique to quantify the uncertainty in item ability estimates. Additionally, we prove the consistency of structure change recognition, ensuring the robustness of the proposed framework. Experiments on both synthetic and real-world datasets demonstrate the practical utility and interpretability of our approach.

## 1 Introduction

Preference-based data, where observations arise from pairwise or groupwise comparisons rather than absolute measurements, is prevalent across various domains. This form of data naturally appears in applications such as economics [1], online recommendations [2], and sports analytics [3]. In addition, the use of preference-based data in reinforcement learning from human feedback (RLHF) has led to significant improvements in the performance of large language models [4]. One major advantage of preference-based data lies in its ease of collection, as it is often more intuitive to express relative preferences rather than assign absolute scores. Many widely used datasets are inherently preference-based, making their effective modeling and analysis a pivotal research focus. To handle such data, the celebrated Bradley-Terry model [5] is widely used for inferring latent preference scores from pairwise comparisons. This model and its extensions have been extensively studied; see [6–9].

Ranking serves as a crucial tool for summarizing preference-based data, providing interpretable outcomes that facilitate decision-making in various fields. It has broad applications, including the evaluation of sports teams [10], institutions [11, 12], recommendation systems [13, 14], financial markets [15, 16], and bioinformatics [17, 18]. By leveraging ranking positions, comparison results enable the identification of top-performing entities [19, 20] while also uncovering underlying trends and patterns [21, 22]. Items to be ranked often possess latent structures due to population homogeneity, which can be reflected in phenomena such as circular comparison results. Moreover, even a slight

---

*Corresponding author.

Code available at https://github.com/nanlu99/RankStruct.

39th Conference on Neural Information Processing Systems (NeurIPS 2025).

modification in comparisons can result in a different rank [23], highlighting the importance of grouped rankings. Grouping similar items can enhance robustness and reduce sensitivity to specific comparisons. For example, group rankings recognize homogeneous entities to improve interpretability and predictive accuracy [10, 24]. In the context of university rankings, Soh [25] argues that minor differences in scores should be ignored and that similar institutions should be assigned to the same group. Given time-varying comparison results, we aim to address the following questions:

• Which items exhibit similar behavior and can be categorized into the same group during a specific period?

• What is the ranking order of these groups at a particular time point?

When considering the temporal dimension, we often encounter situations where item groups evolve over time. For example, in basketball, player trades and coaching changes can significantly affect a team's performance, potentially elevating it to a higher ranking tier. Similarly, the share prices of certain companies may surge due to emerging political, technological, or market factors. These rapidly evolving scenarios underscore the importance of detecting structural change points for long-term analysis. In this context, the term *group change* refers to shifts in group membership over time, which are crucial for accurate analysis in the dynamic grouping problem. In this work, we take a closer look at changes in group structures and aim to address another critical question:

• When does the underlying cluster structure experience significant changes?

There have been several studies on grouping methods for ranking problems in the BT model. Masarotto and Varin [10] first apply the fused lasso penalty to the maximum likelihood estimation. Vana et al. [26] utilize a similar method for journal meta-ranking, and Jeon and Choi [27] extend it to the Luce model. Tian and Shi [28] further consider the problem using the spectral method. However, it is worth noting that all these grouping methods are designed for static situations. In practice, the latent abilities of sports players and institutions may vary over time. Treating data as if it were all collected simultaneously can lead to misleading results. For example, a player in his rising period and another in a declining period may exhibit similar average performances in a game season, but they should not be classified as the same. Hence, this paper concentrates on the simultaneous ranking and grouping problem for the dynamic scenario. Li et al. [3] introduce a segmented static BT model, focusing on detecting the change points of score variation. In contrast, our approach allows scores to vary continuously, and our emphasis lies in recognizing the underlying structure and the changes in clustered groups over time.

We summarize our major contributions as follows.

• **An innovative framework for ranking group recognition.** Though item abilities can be modeled using continuous functions, the ranking positions are discrete functionals of the latent abilities, posing challenges in identifying their structure. To address this, we propose a novel workflow that nests recovering dynamic ranking groups in group change recognition.

• **A generally applicable structure change detection method.** Some works study the clustering problem of different items [10, 26, 27], while few works consider the abrupt changes of item abilities [3]. To the best of our knowledge, we are the first to consider the ranking structure changes for the BT model. We carefully design an integrated objective function, which possesses separable properties, making it permissible to develop an efficient algorithm based on dynamic programming.

• **Theoretical results on recognition consistency and estimator uncertainty.** We characterize conditions of the variability within groups that ensure the consistency recognition of groups and establish the structure recognition consistency. We quantify the uncertainty of item ability estimators using an innovative group inverse technique.

• **Ranking results with enhanced interpretability and improved accuracy.** Our method provides concise ranking results and group change information. The structured ranking results enable the identification of homogeneous items and dynamic group changes, which provide insights into the underlying structure. Simulation results also demonstrate that the grouping method integrates data effectively, yielding improved estimation accuracy.

**Notations** We write $a_n \lesssim b_n$ or $a_n = O(b_n)$ if there exists a constant $c > 0$ such that $a_n \leq cb_n$ for all $n$. We denote $a_n \asymp b_n$ if $b_n \lesssim a_n$ and $a_n \lesssim b_n$. Besides, we write $a_n = o(b_n)$ or $a_n \ll b_n$ if $\lim_{n\to\infty} a_n/b_n = 0$. We denote by $[n] = 1, \ldots, n$ for any positive integer $n$. We let $\mathbf{I}$ represent

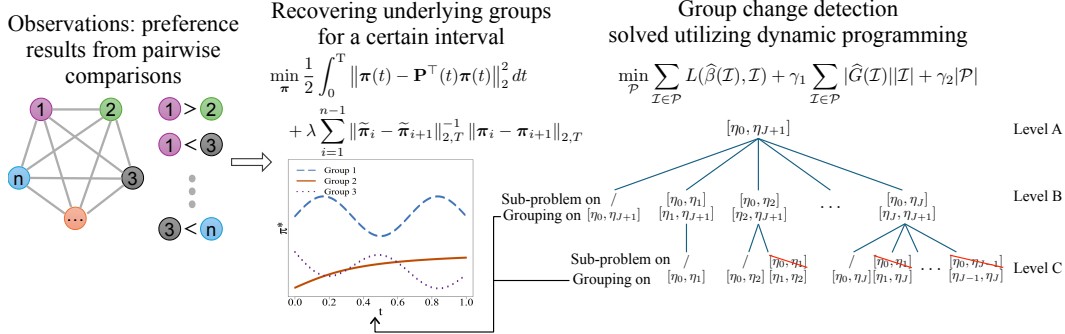

Figure 1: Workflow of the proposed method for dynamic ranking structure recognition. The left panel depicts the data format, representing preference outcomes obtained from pairwise comparisons. The middle panel illustrates the score evolution of each group over a given interval. The right panel shows the structure detection procedure, where: (1) each node can be decomposed into subproblems represented by its child nodes; (2) each child node corresponds to a subproblem and a grouping problem on intervals; (3) subproblems at level C reuse results from nodes preceding their parent node at level B, thereby avoiding redundant computation.

the identity matrix, and let $e_n$ be an $n \times 1$ vector with each element equal to 1. $\mathbf{0}$ represents the vector or matrix composed entirely of zeros. For a vector $v$, $\|v\|_2$ denotes the $\ell_2$-norm. We let $\|f\|_{2,T} = (\int_0^T f(t)^2 dt)^{1/2}$, where $f(t)$ is a function on $[0, T]$.

## 2 Ranking Structure Recognition

### 2.1 Dynamic Bradley-Terry Model

In a dynamic scenario, we observe pairwise comparison results among $n$ items, denoted as $\mathcal{Y} = \{y_{ij}(t_k), i, j \in [n], t_k \in T_{ij}\}$. The scalar $y_{ij}(t)$ represent the comparison result at time point $t$ between items $i$ and $j$, where $y_{ij}(t) = 1$ represents that item $i$ wins over item $j$, and $y_{ij}(t) = 0$ indicates the opposite. We assume that the elements of $\mathcal{Y}$ are independent, and the comparison time $T_{ij}$ of $(i, j)$ pair is uniformly distributed over $[0, T]$. The Bradley-Terry model assigns positive scores $\boldsymbol{\pi}^*(t) = (\pi_1^*(t), \pi_2^*(t), \ldots, \pi_n^*(t))^\top$ to items, and presumes $y_{ij}(t) \sim$ Bernoulli$(y_{ij}^*(t))$, where $y_{ij}^*(t) = \pi_j^*(t)/(\pi_i^*(t) + \pi_j^*(t))$ [5]. Intuitively, taking $\pi_i^*(t)$ as the ability of item $i$, $y_{ij}^*(t)$ represents the winning rate of item $i$. Notice that the BT model is invariant under the scaling of the scores, so we set $\sum_{i=1}^n \pi_i^*(t) = 1$ for all $t \in [0, T]$ to obtain a unique representation.

A straightforward approach to estimate the BT model is the maximum likelihood estimator [29, 30]. In pursuit of a computationally efficient solution, we opt for the spectral-based solver [6, 22]. Negahban et al. [6] present an insightful perspective of the spectral method, establishing a connection between the pairwise comparison results and the transition of a Markov chain. Specifically, by letting the nodes of a graph represent the items, and assigning the transformation probability from node $i$ to $j$ based on the frequency of item $i$ losing to $j$, it is proven that the stationary distribution of this random walk corresponds to the items' abilities $\boldsymbol{\pi}^*$. For a more detailed understanding, we recommend referring to Negahban et al. [6].

We adopt the kernel-based estimator. Let $K_h(t, s) = \frac{1}{h} K\left(\frac{t-s}{h}\right)$, where $K(\cdot)$ is the kernel function and $h$ is the bandwidth. The transformation probability matrix $\mathbf{P}(t)$ is formulated as

$$\mathbf{P}_{ij}(t) = \begin{cases} \frac{1}{n} \frac{\sum_{t_k \in T_{ij}} y_{ij}(t_k) K_h(t, t_k)}{\sum_{t_k \in T_{ij}} K_h(t, t_k)} & \text{if } i \neq j, \\ 1 - \sum_{s \neq i} \mathbf{P}_{is}(t) & \text{if } i = j. \end{cases}$$

Then we have the consistent estimator $\widetilde{\boldsymbol{\pi}}(t)$, which is a stationary distribution of the Markov chain deduced by the stochastic matrix $\mathbf{P}$ [22].

## 2.2 Recognition of Dynamic Ranking Groups

We first consider recovering the ranking groups for a given interval in this section, which is a cornerstone for analyzing the changes of ranking groups for a relatively longer time period as discussed in Section 2.3.

Let $B$ denote the number of groups. We represent these groups as $G = \{G_1, G_2, \ldots, G_B\}$, forming a partition of the set $[n]$. The items within the same group possess similar scores, while those from different groups have significant score differences. Formally, we have

$$\delta_1 := \min_{\substack{k,l \in [B] \\ k \neq l}} \min_{\substack{i \in G_k \\ j \in G_l}} \|\boldsymbol{\pi}_i^* - \boldsymbol{\pi}_j^*\|_{2,T} \gg \max_{k \in [B]} \max_{i,j \in G_k} \|\boldsymbol{\pi}_i^* - \boldsymbol{\pi}_j^*\|_{2,T}.$$

Without loss of generality, we assume that $\max\{i : i \in G_k\} < \min\{i : i \in G_l\}$ for $k < l$.

We then recover the partition of items and present the score estimations simultaneously. Since any finite-state time-homogeneous Markov chain has at least one stationary distribution, we rewrite the estimator $\widetilde{\boldsymbol{\pi}}(t_0)$ as the solution of the optimization problem, $\min_{\widetilde{\boldsymbol{\pi}}(t_0)} \|\widetilde{\boldsymbol{\pi}}(t_0) - \mathbf{P}^\top(t_0)\widetilde{\boldsymbol{\pi}}(t_0)\|_2$ such that $\sum_{i=1}^n \widetilde{\boldsymbol{\pi}}_i(t_0) = 1$. Setting $\lambda$ as a tuning parameter, we consider the following objective function.

$$\min_{\boldsymbol{\pi}} \quad \frac{1}{2} \int_0^T \|\boldsymbol{\pi}(t) - \mathbf{P}^\top(t)\boldsymbol{\pi}(t)\|_2^2 \, dt + \lambda \sum_{i=1}^{n-1} \|\widetilde{\boldsymbol{\pi}}_i - \widetilde{\boldsymbol{\pi}}_{i+1}\|_{2,T}^{-1} \|\boldsymbol{\pi}_i - \boldsymbol{\pi}_{i+1}\|_{2,T}$$

$$\text{s.t.} \quad \sum_{i=1}^n \boldsymbol{\pi}_i(t_k) = 1, \ k = 1, 2, \ldots, m. \tag{2.1}$$

Let $t_1, t_2, \ldots, t_m$ be $m$ equidistant time points in $[0, T]$. We use the symbol with an item subscript, such as $\boldsymbol{\pi}_i$, to represent the vector corresponding to the $m$ time points $(\boldsymbol{\pi}_i(t_1), \boldsymbol{\pi}_i(t_2), \ldots, \boldsymbol{\pi}_i(t_m))^\top$. Here, the parameter $m$ is allowed to approach infinity, allowing for the approximation of the integral. To efficiently address the constrained optimization problem, we employ a technical transformation, leading us to an unconstrained form with a well-developed optimization algorithm. Define the $n \times n$ matrix

$$\mathbf{Q}_{ij} = \begin{cases} 1 & \text{if } i = j \text{ or } i = n, \\ -1 & \text{if } i < n \text{ and } j = i + 1, \\ 0 & \text{otherwise.} \end{cases}$$

Let $\boldsymbol{\theta}(t) = \mathbf{Q}(\boldsymbol{\pi}(t) - \frac{1}{n}\boldsymbol{e}_n)$ and $\widetilde{\boldsymbol{\theta}} = \mathbf{Q}(\widetilde{\boldsymbol{\pi}}(t) - \frac{1}{n}\boldsymbol{e}_n)$. Let $\boldsymbol{\theta}(t) = (\boldsymbol{\theta}_1(t), \ldots, \boldsymbol{\theta}_{n-1}(t))^\top$ and $\boldsymbol{\theta} = (\boldsymbol{\theta}(t_1)^\top, \ldots, \boldsymbol{\theta}(t_m)^\top)^\top$. Let $\boldsymbol{\theta}^*$ be the corresponding true value (with $\boldsymbol{\pi}$ substituted by $\boldsymbol{\pi}^*$), and $\widetilde{\boldsymbol{\theta}}$ denote the counterpart induced by $\widetilde{\boldsymbol{\pi}}$. We define $\mathbf{X}(t) = (\mathbf{P}^\top(t) - \mathbf{I})\mathbf{Q}^{-1}$ and $\boldsymbol{Y}(t) = \frac{1}{n}(\mathbf{I} - \mathbf{P}^\top(t))\boldsymbol{e}_n$. Then we have the optimization problem (2.1) reformulated as

$$\min_{\boldsymbol{\theta}} \frac{1}{2}\|\boldsymbol{Y} - \mathbf{X}\boldsymbol{\theta}\|_2^2 + \lambda \sum_{i=1}^{n-1} \|\widetilde{\boldsymbol{\theta}}_i\|_2^{-1} \|\boldsymbol{\theta}_i\|_2, \tag{2.2}$$

where $\mathbf{X}$ is the $mn \times m(n-1)$ matrix $\text{diag}(\mathbf{X}_{-1}(t_1), \ldots, \mathbf{X}_{-1}(t_m))$, $\mathbf{X}_{-1}(t)$ is the matrix $\mathbf{X}(t)$ with its last column removed. $\boldsymbol{Y}$ represents the $mn \times 1$ vector, $(\boldsymbol{Y}(t_1), \ldots, \boldsymbol{Y}(t_m))^\top$. This transformation directly eliminates the constraints, reducing the optimization objective to a standard adaptive group lasso problem, which possesses efficient solutions. Having obtained the solution $\widehat{\boldsymbol{\theta}}$, we can calculate $\widehat{\boldsymbol{\pi}}(t) = \mathbf{Q}^{-1}(\widehat{\boldsymbol{\theta}}(t)^\top, 0)^\top + \frac{1}{n}\boldsymbol{e}_n$. Let $\mathcal{S} = \{i : \boldsymbol{\theta}_i^* \neq \mathbf{0}\}$, $\widehat{\mathcal{S}} = \{i : \widehat{\boldsymbol{\theta}}_i \neq \mathbf{0}\}$ and $\widehat{B} = |\widehat{\mathcal{S}}| + 1$. We use $\widetilde{\mathcal{S}}$ to denote the estimated partition points of different groups. Specifically, we let $\widetilde{\mathcal{S}} = \{0\} \cup \widehat{\mathcal{S}} \cup \{n\}$. Without loss of generality, we assume $\widetilde{\mathcal{S}}$ is arranged in ascending order (if not, we simply reorder $\widehat{\mathcal{S}}$), and $\widetilde{\mathcal{S}}_i$ denotes the $i$-th element of $\widetilde{\mathcal{S}}$. The group estimation $\widehat{G} = \{\widehat{G}_1, \widehat{G}_2, \ldots, \widehat{G}_{\widehat{B}}\}$ is obtained by $\widehat{G}_k = \{i : \widetilde{\mathcal{S}}_{k-1} < i \leq \widetilde{\mathcal{S}}_k\}$.

**Remark 2.1.** We note that our approach and the theoretical justification presented below do not rigidly require that all items within a group possess identical scores. Instead, we establish a framework in which items within a group exhibit similar behavior, rendering practically flexibility.

**Remark 2.2.** Though we originally have the fused term among different items as in (2.1), the optimization objective has the same expression as the adaptive group lasso of a linear regression model in (2.2). It is worth noting that the similarity is somehow superficial since the design matrix $\mathbf{X}$ is no longer deterministic, posing difficulties for theoretical analysis.

To mitigate the issue of shrinkage in large coefficients resulting from the penalization term, a widely utilized approach is the refit procedure [31, 32]. This method entails re-estimating the coefficients after identifying the underlying structure. Upon obtaining the group estimation $\widehat{G}$, we employ the refit strategy in the following manner. For $i, j \in [\widehat{B}]$, define

$$
\mathbf{P}_{\widehat{G}ij}(t) = \begin{cases} \frac{1}{\widehat{B}} \frac{\sum_{l_1 \in \hat{G}_i} \sum_{l_2 \in \hat{G}_j} \sum_{t_k \in T_{l_1 l_2}} y_{l_1 l_2}(t_k) K_h(t, t_k)}{\sum_{l_1 \in \hat{G}_i} \sum_{l_2 \in \hat{G}_j} \sum_{t_k \in T_{l_1 l_2}} K_h(t, t_k)}, & \text{if } i \neq j; \\ 1 - \sum_{s \neq i} \mathbf{P}_{\widehat{G}is}(t), & \text{if } i = j. \end{cases}
$$

We can obtain the stationary distribution $\widehat{\boldsymbol{\pi}}_{\widehat{G}} = (\widehat{\boldsymbol{\pi}}_{\widehat{G}_i})_{i \in [\widehat{B}]}$ of $\mathbf{P}_{\widehat{G}}(t)$. Note that we have assumed the score summation of $n$ items to be 1 to eliminate the non-uniqueness caused by rescaling. Therefore, the refit estimator for each item is

$$
\widehat{\boldsymbol{\pi}}_i^{rf}(t) = \frac{\widehat{\boldsymbol{\pi}}_{\widehat{G}_l}(t)}{\sum_{k \in [|\widehat{G}|]} |\widehat{G}_k| \widehat{\boldsymbol{\pi}}_{\widehat{G}_k}(t)}, \, i \in \widehat{G}_l. \tag{2.3}
$$

**Remark 2.3.** We use the absolute group size for normalization because, as stated in Section 2.1, we impose the constraint that the sum of item abilities equals 1, i.e., $\sum_{i=1}^n \pi_i^* = 1$, to ensure a unique representation. Since only the ratio between scores matters in the BT model, this constraint guarantees identifiability. Furthermore, when recovering the original scores after refitting, we still expect the normalized scores to satisfy $\sum_{i=1}^n \widehat{\pi}_i^{rf} = 1$. At the same time, we need to preserve the score ratios between items from different groups, meaning that for $i \in \widehat{G}_l$ and $j \in \widehat{G}_k$, $\widehat{\pi}_i^{rf}/(\widehat{\pi}_i^{rf} + \widehat{\pi}_j^{rf}) = \widehat{\pi}_{\widehat{G}_l}/(\widehat{\pi}_{\widehat{G}_l} + \widehat{\pi}_{\widehat{G}_k})$. To satisfy both conditions simultaneously, the normalization in equation (2.3) is scaled by the absolute group size $|\widehat{G}_k|$.

**Remark 2.4.** The refit strategy is an optional part. Treating the comparison result of items in a group as one actually compensates for more information, especially in cases where $n$ is large and $Mh$ is small. We also point out that refitting induces better performance, as indicated in simulations.

## 2.3 Recognition of Group Changes

We then focus on detecting the change points of latent clusters over an extended period. Consider a scenario with time-correlated observations occurring within the interval $[0, V]$. Still consider $n$ entities whose structure needs to be determined. There are $J + 1$ phases, where the items' groups remain unchanged within each phase and differ between adjacent phases. In a more formal mathematical form, let $\boldsymbol{z}(t) = (\boldsymbol{z}_1(t), \ldots, \boldsymbol{z}_n(t))^\top$ represent the latent group of items at the time point $t$. There are $J + 2$ points $0 = \eta_0 < \ldots < \eta_{J+1} = V$ such that $\boldsymbol{z}(t) \neq \boldsymbol{z}(s)$ for $\eta_{i-1} < t < \eta_i < s < \eta_{i+1}$, $i \in [J]$ and $\boldsymbol{z}(t) = \boldsymbol{z}(s)$ for $\eta_{i-1} < t, s < \eta_i$, $i \in [J + 1]$. The unobservable structure change points $\{\eta_i\}_{i \in [J]}$ belong to a preset candidate set $\{\xi_i\}_{i \in [U]}$. Without loss of generality, let $\{\xi_i\}_{i \in [U]}$ be in increasing order. In practice, the candidate set may be selected based on practical considerations, such as dividing points between seasons in sports games or uniformly distributed time points.

To detect changes in underlying groups, it is necessary to employ clustering methods within a subinterval $\mathcal{I} \subset [0, V]$. We utilize the clustering method proposed in Section 2.2 for dynamic ranking, by simply substituting $[0, T]$ with $\mathcal{I}$. Let $\widehat{G}(\mathcal{I})$ represent the estimated group corresponding to the true structure $G(\mathcal{I})$. With a slight abuse of notations, let $\widehat{\beta}(\mathcal{I})$ denote the model parameter estimations $\{\widehat{\boldsymbol{\pi}}_i^{rf}(t), i \in [n], t \in \mathcal{I}\}$, and let $\beta(\mathcal{I}) = \{\beta(t), t \in \mathcal{I}\}$ be the corresponding true values. Define $\bar{y}_{ij}(t) = \frac{\sum_{k=1}^M y_{ij}(t_k) K_h(t, t_k)}{\sum_{k=1}^M K_h(t, t_k)}$. We introduce the negative log-likelihood function for $\boldsymbol{\pi} = (\boldsymbol{\pi}_1, \ldots, \boldsymbol{\pi}_n)^\top$ at a time point $t$,

$$
l(\boldsymbol{\pi}, t) = -\frac{2}{n(n-1)} \sum_{(i,j):i \neq j} \bar{y}_{ij}(t) \log(\frac{\boldsymbol{\pi}_j}{\boldsymbol{\pi}_i + \boldsymbol{\pi}_j}), \tag{2.4}
$$

which is a natural extension of the static case. Define the function $L(\widehat{\beta}(\mathcal{I}), \mathcal{I}) = \int_{t \in \mathcal{I}} l(\widehat{\beta}(t), t) \, dt$ to measure the discrepancy between observed samples and the values expected under the grouping model.

Let $\mathcal{P}$ represents $\{[s_0, s_1), [s_1, s_2), \ldots, [s_p, s_{p+1}]\}$, with $s_0 = 0$, $s_{p+1} = V$ and $\{s_i\}_{i \in [p]} \subset \{\xi_i\}_{i \in [U]}$ being a list of increasing points. We use $|\widehat{G}(\mathcal{I})|$ to represent the estimated group number

and $|\mathcal{I}|$ to denote the interval length. We recover the change points of structures by considering the objective function:

$$\min_{\mathcal{P}} \sum_{\mathcal{I} \in \mathcal{P}} L(\widehat{\beta}(\mathcal{I}), \mathcal{I}) + \gamma_1 \sum_{\mathcal{I} \in \mathcal{P}} |\widehat{G}(\mathcal{I})||\mathcal{I}| + \gamma_2 |\mathcal{P}|. \tag{2.5}$$

Intuitively, the first term evaluates the goodness of fit for the parameters, the second term is the penalty of groups and the last term imposes a penalty on phase changes.

We provide a brief clarification that the framework exhibits versatility. It is not confined to the ranking problem but can be applied to the general detection of group changes. As long as a clustering method designed for subintervals is provided, the framework can effectively perform the structure change detection. Specifically, it relies on $\widehat{G}(\mathcal{I})$ to present the grouping results and $L(\widehat{\beta}(\mathcal{I}), \mathcal{I})$ to assess the goodness of fit of the grouping method. Besides negative log-likelihood functions, $l(\cdot)$ can be residuals or other measurements, determined by the specific problem.

Note that the optimization objective (2.5) exhibits separability with respect to time and features an optimal substructure property. Specifically, the objective has an additive form across time, which allows it to be decomposed into independent subproblems separable over time. Moreover, the optimal solution to the overall problem can be constructed from the optimal solutions of its subproblems. These two properties enable the objective to be optimized recursively, forming the basis of an efficient dynamic programming solution. We can address the combinatorial problem using Algorithm 1, which provides an efficient method to estimate $\mathcal{R} = \{\widehat{s}_i\}_{i \in [\widehat{J}]}$.

---

**Algorithm 1** Structure Change Detection

**Require:** Observed data $\mathcal{Y}$, tuning parameters $\gamma_1, \gamma_2$.
**Ensure:** Change points estimation $\mathcal{R}$.
$\quad R = \emptyset, \boldsymbol{a} = -\boldsymbol{e}_{U+1}, b = (\infty, \ldots, \infty) \in \mathbb{R}^{U+1}, b_0 = 0, \xi_0 = 0, \xi_{U+1} = V$.
$\quad$**for** $r$ from 1 to $U + 1$ **do**
$\quad\quad$**for** $l$ from 0 to $r - 1$ **do**
$\quad\quad\quad b \leftarrow b_l + L(\widehat{\beta}(\mathcal{I}), \mathcal{I}) + \gamma_1 |\widehat{G}(\mathcal{I})||\mathcal{I}| + \gamma_2$, where $\mathcal{I} = [\xi_l, \xi_r]$.
$\quad\quad\quad$**if** $b < b_r$ **then**
$\quad\quad\quad\quad b_r \leftarrow b; \boldsymbol{a}_r \leftarrow l$.
$\quad k \leftarrow U + 1$
$\quad$**while** $k > 0$ **do**
$\quad\quad d \leftarrow \boldsymbol{a}_k; \mathcal{R} = \mathcal{R} \cup \{\xi_d\}; k \leftarrow d$.
$\quad$**return** $\mathcal{R}$

---

## 3 Statistical Learning Theory

### 3.1 Consistency of Ranking Group Estimation

In this section, we present theoretical guarantees for our estimator. Specifically, we show that the probability of correctly identifying the underlying group structure approaches one as the sample size increases. We refer to this property as *group consistency*, a desirable feature that supports the reliability of the proposed estimation method. To ensure the theoretical results, we introduce the following assumptions.

**Assumption 3.1.** $\sup_{t \in [0,T]} \frac{\max_i \boldsymbol{\pi}_i^*(t)}{\min_i \boldsymbol{\pi}_i^*(t)} \leq \kappa$, where $\kappa > 0$ is a constant. $\boldsymbol{\pi}_i^*(t)$ is three times continuously differentiable, $i \in [n]$.

**Assumption 3.2.** The kernel function is symmetric, nonnegative, and satisfies $\int_{-\infty}^{\infty} K(v) \, dv = 1$ and $\int_{-\infty}^{\infty} v^2 K(v) \, dv < \infty$.

Assumptions 3.1 and 3.2 are commonly used in the BT model and kernel methods [30, 19]. We let $|T_{ij}| = M$ for $i, j \in [n]$. We note that our method applies to the case where the number of comparisons may vary over time, and we assume a constant number of comparisons only for the simplicity of presentation. Recall that $\mathcal{S} = \{i : \boldsymbol{\theta}_i^* \neq \mathbf{0}\}$. Let $\delta_2 \geq 0$ be a constant such that

$|\boldsymbol{\theta}_i^*(t)| \leq \delta_2, \forall t \in T, i \in \mathcal{S}^c$. Define $n_i$ as the number of items in $G_i$, and let $r_i = n_i/n$. Assume $r_i \asymp 1/B$, $i \in [B]$. Let $\delta = \sqrt{\frac{\log(nM)}{n^3 Mh}}$, which denotes the uniform convergence rate of the KRC estimator [19].

**Theorem 3.3.** Let Assumptions 3.1 and 3.2 hold. When $Mh \to \infty$, $n \to \infty$ and $nMh^5 \to 0$, if

1. $\max\{\delta, \frac{1}{m}, \sqrt{\frac{B}{n^3 Mh}}\} = o(\delta_1)$ and $\delta_2 = o(\sqrt{\frac{1+\cos\frac{(n-B)\pi}{n-B+1}}{B(n-B)}\frac{1}{n^2 Mh}})$; 2. $\sqrt{\frac{m^2 B^3}{nMh}}\widetilde{\delta} \ll \lambda \lesssim \frac{\delta_1\sqrt{m}}{B\sqrt{nMh}}$,

where $\widetilde{\delta} = \max\{\delta, \delta_2\}$, then we have $\mathbf{P}(\widehat{G} = G) \to 1$.

We have established the group consistency property. The following two remarks clarify the conditions of the theorem and highlight the distinct features of our theoretical analysis.

**Remark 3.4.** The first condition characterizes the requirement for $\delta_1$ to recognize the differences among groups without being impeded by estimation errors, while the requirement of $\delta_2$ limits the variation within each group to ensure accurate item ability estimation. The second condition requires the appropriate order of the penalized parameter $\lambda$. The term $1/m$ denotes the order of integral approximation error for $\|\widetilde{\boldsymbol{\theta}}_i\|_2$ and is not essential. If the midpoint approximation is replaced by the trapezoidal rule, then $1/m$ is replaced by $1/m^2$.

**Remark 3.5.** Unlike standard linear regression, the design matrix $\mathbf{X}$ in this context is derived from a series of transformations on the observed data $y$. This introduces challenges for theoretical analysis. Fortunately, $\mathbf{P}(t)$ is an approximation of a reversible Markov transition matrix $\mathbf{P}^*(t)$ (see Section C.1), where

$$\mathbf{P}_{ij}^*(t) = \begin{cases} \frac{1}{n}\frac{\pi_j^*(t)}{\pi_i^*(t)+\pi_j^*(t)} & \text{if } i \neq j; \\ 1 - \sum_{s\neq i}\mathbf{P}_{is}^*(t) & \text{if } i = j. \end{cases}$$

That plays an important role in deducing the properties of $\mathbf{X}$ and $\mathbf{P}$ and facilitates the establishment of theoretical guarantees.

### 3.2 Asymptotic Distribution of Item Ability Estimates

In this section, we discuss uncertainty quantification, that is, the asymptotic distribution of the item ability estimators. A well-characterized uncertainty quantification enables statistical inference tasks such as hypothesis testing and helps assess the reliability of the estimators.

Selecting one representative item from each group, $i_1 \in G_1, i_2 \in G_2, \ldots, i_B \in G_B$, let $\boldsymbol{\pi}_G^*(t) = (\pi_{G_1}^*(t), \ldots, \pi_{G_B}^*(t))^\top = (\pi_{i_1}^*(t), \ldots, \pi_{i_B}^*(t))^\top / \sum_{k=1}^B \pi_{i_k}^*(t)$ and $\widehat{\boldsymbol{\pi}}_G(t) = (\widehat{\pi}_{i_1}^{rf}(t), \ldots, \widehat{\pi}_{i_B}^{rf}(t))^\top / \sum_{k=1}^B \widehat{\pi}_{i_k}^{rf}(t)$. We observe that $\boldsymbol{\pi}_G^*$ is the stationary distribution of the $B \times B$ matrix $\mathbf{P}_G^*(t)$, where

$$\mathbf{P}_{Gij}^*(t) = \begin{cases} \frac{\pi_{Gj}^*(t)}{\pi_{Gi}^*(t)+\pi_{Gj}^*(t)}, & \text{if } i \neq j; \\ 1 - \sum_{s\neq i}\mathbf{P}_{Gis}^*(t), & \text{if } i = j. \end{cases}$$

Set $\mathbf{A}^{\#}(t)$ as the group inverse of $\mathbf{I} - \mathbf{P}_G^*(t)$ (see the definition of group inverse in Kirkland and Neumann [33]). We have the following result.

**Theorem 3.6.** Under the conditions of Theorem 3.3, if $\delta_2 = o(\frac{1}{\sqrt{n^4 Mh}})$, for a fixed $B$ and any $t \in (0, 1)$, we have

$$\sqrt{n^2 Mh}(\widehat{\boldsymbol{\pi}}_G(t) - \boldsymbol{\pi}_G^*(t)) \xrightarrow{\mathcal{D}} N(0, \boldsymbol{\Gamma}(t)\boldsymbol{\Lambda}(t)\boldsymbol{\Gamma}(t)^\top),$$

where $\boldsymbol{\Lambda}(t)$ is a $\frac{B(B-1)}{2}$ diagonal matrix with $\boldsymbol{\Lambda}_{kl,kl}(t) = \frac{1}{r_k r_l}\frac{\pi_{G_k}^*(t)\pi_{G_l}^*(t)}{(\pi_{G_k}^*(t)+\pi_{G_l}^*(t))^2}\int K^2(v)dv$, and $\boldsymbol{\Gamma}(t)$ is a $B \times \frac{B(B-1)}{2}$ matrix with $\boldsymbol{\Gamma}_{i,kl}(t) = (\mathbf{A}_{li}^{\#}(t) - \mathbf{A}_{ki}^{\#}(t))\frac{(\pi_{G_k}^*(t)+\pi_{G_l}^*(t))}{B}$, $1 \leq i \leq B$, $1 \leq k < l \leq B$.

### 3.3 Consistency of Group Changes Detection

In this section, we focus on *structure recognition consistency*, which refers to the property that the probability of correctly identifying group structure change points approaches one as the sample size increases. This property is important as it ensures reliable detection of structural changes. We first provide a general analysis for arbitrary grouping methods in Theorem 3.9, and then specialize the discussion to the dynamic ranking setting in Corollary 3.10.

To guarantee the correctness of the estimated change points, we impose assumptions regarding the grouping accuracy within a given time interval $\mathcal{I}$.

**Assumption 3.7.** As sample size tends to infinity, we have $P(\widehat{G}(\mathcal{I}) = G(\mathcal{I})) \to 1$.

**Assumption 3.8.** $\frac{1}{|\mathcal{I}|}|L(\beta(\mathcal{I}), \mathcal{I}) - L(\widehat{\beta}(\mathcal{I}), \mathcal{I})| = O_p(\delta_3)$, where $\delta_3 \to 0$ as sample size tends to infinity.

Assumption 3.7 is intended to ensure the accurate recovery of groups. With a consistent estimator $\widehat{\beta}(\mathcal{I})$, Assumption 3.8 can be satisfied by incorporating a sufficiently smooth $l(\cdot)$. We show in Corollary 3.10 that our method in Section 2.2 is capable of satisfying these assumptions.

**Theorem 3.9.** Under Assumptions 3.7 and 3.8, if $\gamma_1 > \gamma_2$, and $\delta_3 = o(\gamma_2)$, then $P(\{\widehat{s}_i\}_{i=1}^{\widehat{J}} = \{\eta_i\}_{i=1}^{J}) \to 1$ with sample size tending to infinity.

Intuitively, the order of $\gamma_2$ should be larger than that of $\delta_3$ to ensure efficient penalty and $\gamma_1$ is supposed to be larger than $\gamma_2$ to avoid missing change points. Based on Theorems 3.3 and 3.9, we have the following consistency guarantee for the ranking group change detection.

**Corollary 3.10.** Under the conditions of Theorem 3.3, if $\gamma_1 > \gamma_2$ and $\sqrt{\frac{1}{nMh}} = o(\gamma_2)$, we have $P(\{\widehat{s}_i\}_{i=1}^{\widehat{J}} = \{\eta_i\}_{i=1}^{J}) \to 1$ as $n \to \infty$.

## 4 Computational Experiments

### 4.1 Recognition of Dynamic Ranking Groups

We evaluate the results using the Kendall $\tau$ coefficient and the mean squared error (MSE) between the estimators $(\widehat{\boldsymbol{\pi}}(t), \widehat{\boldsymbol{\pi}}^{rf}(t))$ and $\boldsymbol{\pi}^*(t)$ to assess the estimation accuracies of rank and value. We employ sensitivity and specificity to gauge group accuracy. Specifically, sensitivity represents the proportion of correctly identified pairs within the same group, while specificity calculates the percentage correctly distinguished between different groups. We compare our method (without and with refit strategy) with the static clustering method Group Rank Centrality (GRC) [28] and the original estimator Kernel Rank Centrality (KRC). All experiments are conducted on a machine with an 11th Gen Intel(R) Core(TM) i5-1135G7 CPU and 16GB RAM. We utilize R pacakge sparsegl [34] for analysis. We consider two different experimental settings. Detailed configurations and parameter choices are provided in Section B.1, and the results are summarized in Table 1.

Table 1: Simulation results for simultaneously grouping and ranking.

| Kendall $\tau$ | | | | MSE | | | | Sensitivity | | Specificity | |
|---|---|---|---|---|---|---|---|---|---|---|---|
| Ours | Ours (refit) | GRC | KRC | Ours | Ours (refit) | GRC | KRC | Ours | GRC | Ours | GRC |
| Setting 1 (n=20, Mh=5) | | | | | | | | | | | |
| 0.9998 | 0.9998 | 0.9999 | 0.8330 | 0.0586 | 0.0466 | 0.1083 | 0.1277 | 99.95% | 99.97% | 99.99% | 100.00% |
| Setting 1 (n=50, Mh=10) | | | | | | | | | | | |
| 1.0000 | 1.0000 | 1.0000 | 0.8207 | 0.0421 | 0.0375 | 0.1079 | 0.0684 | 100.00% | 100.00% | 100.00% | 100.00% |
| Setting 2 (n=20, Mh=5) | | | | | | | | | | | |
| 0.9416 | 0.9484 | 0.7977 | 0.7565 | 0.0494 | 0.0359 | 0.2168 | 0.1208 | 100.00% | 100.00% | 100.00% | 63.64% |
| Setting 2 (n=50, Mh=10) | | | | | | | | | | | |
| 0.9644 | 0.9683 | 0.7974 | 0.7742 | 0.0291 | 0.0250 | 0.2167 | 0.0606 | 100.00% | 100.00% | 100.00% | 63.71% |

Since the refit strategy is based on identified groups, the results of non-refit and refit estimators exhibit the same sensitivity and specificity values. The sensitivity and specificity of KRC are not listed as it does not exhibit a grouping effect. It can be observed that the refit estimator performs slightly better than the non-refit one, showing a larger Kendall $\tau$ and a smaller MSE. The Kendall $\tau$ of our method approaches one with increasing sample size, surpassing the values of the other two methods. The results of specificity highlight the necessity of a dynamic setting. It is evident that GRC cannot distinguish some groups in the second setting. Comparing the results of both settings, the Kendall $\tau$ and MSE of our method are superior to those of KRC. This suggests that our method effectively captures group information, yielding better estimation accuracy.

## 4.2 Recognition of Group Changes

Due to the lack of established methods for detecting dynamic ranking structural changes, we compare our method with a naive baseline that groups items within each interval between consecutive candidate change points. A dividing point is identified as a structural change point if the groupings in its adjacent intervals differ. We evaluate the experiment results using two criteria: the number of change points and the Hausdorff distance (H-dist) between the actual and estimated sets of structural change points. Detailed configurations and parameter choices are provided in Section B.2. We summarize the results for two different settings as follows.

Table 2: Simulation results for structural change detection.

| | Ours | | | | Naive | | | |
|---|---|---|---|---|---|---|---|---|
| | H-dist | $\widehat{J} < J$ | $\widehat{J} = J$ | $\widehat{J} > J$ | H-dist | $\widehat{J} < J$ | $\widehat{J} = J$ | $\widehat{J} > J$ |
| Setting 1 | | | | | | | | |
| Mh=2 | 0.0051 | 0.0% | 97.4% | 2.6% | 0.2084 | 0.0% | 0.0% | 100.0% |
| Mh=4 | 0.0004 | 0.0% | 99.8% | 0.2% | 0.1800 | 0.0% | 0.0% | 100.0% |
| Mh=10 | 0.0000 | 0.0% | 100.0% | 0.0% | 0.1333 | 0.0% | 0.8% | 99.2% |
| Setting 2 | | | | | | | | |
| Mh=4 | 0.0492 | 0.0% | 80.4% | 19.6% | 0.3258 | 0.0% | 0.0% | 100.0% |
| Mh=10 | 0.0076 | 0.0% | 97.0% | 3.0% | 0.2842 | 0.0% | 0.4% | 99.6% |
| Mh=20 | 0.0004 | 0.0% | 99.8% | 0.2% | 0.2434 | 0.0% | 3.6% | 96.4% |

Table 2 shows that the estimated change points quickly converge to the true values as the amount of observed data increases. Compared to the naive approach, our method requires significantly fewer samples to recover the true underlying structure, demonstrating the effectiveness of the proposed objective function.

## 5 Empirical Analysis: Ranking Structure Recognition of NBA Teams

We analyze NBA regular season data from the 2014-2015 season to the 2018-2019 season[2]. The candidate structure change points correspond to the season transitions and trade deadlines each season. These trade deadlines typically fall around February 20th each year and are denoted as 'TradeDDL'. We identify two structure change points: the 2015–2016 trade deadline and the end of the 2016–2017 season, with results shown in Figure 2. For each resulting phase, we plot team win rates: alphabetically ordered on the left and ordered groups on the right. Black lines separate distinct groups. The left plot appears random, while the right displays a gradient from dark to light colors, moving from the top left to the bottom right. Items within each group exhibit similar behavior, as reflected by the color uniformity within each block, supporting the validity of the detected structure. Further details are provided in Section B.4.

Figure 3 displays the estimation of teams' strengths using direct estimation and our ranking structure recognition method. The figure illustrates that our method provides a concise result regarding the structure of teams and the team strengths in each group.

---

[2]https://www.nba.com/games

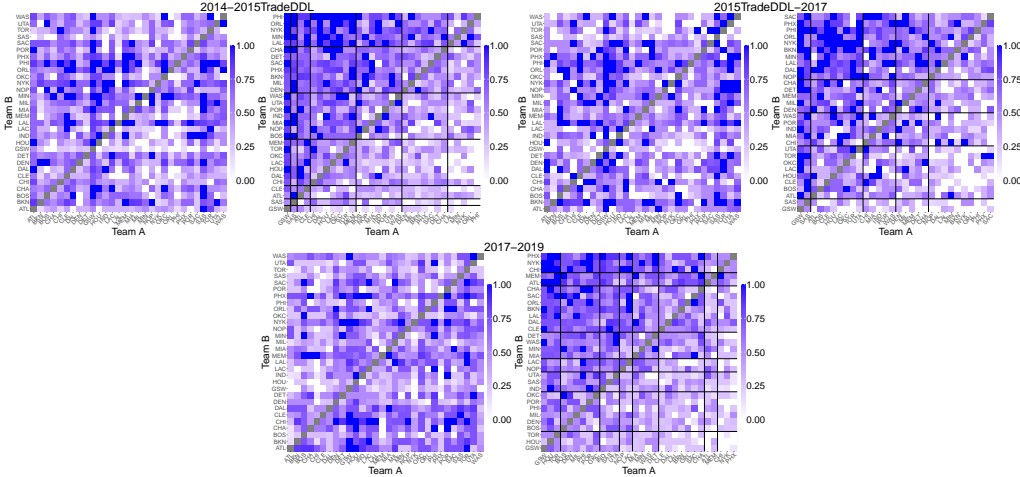

Figure 2: The winning percentage of Team A over Team B.

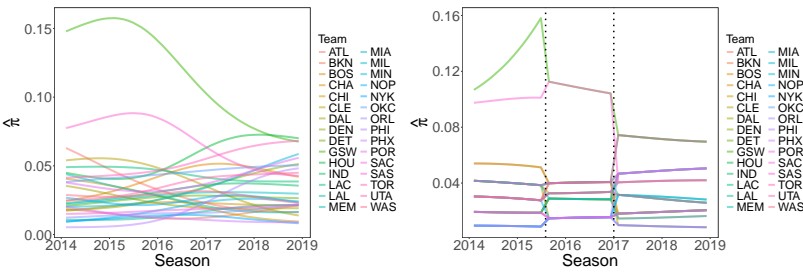

Figure 3: Estimation of team strengths using KRC (left) and our method (right).

## 6 Conclusion

We present a novel approach that simultaneously performs grouping and ranking based on time-varying comparisons. This offers an innovative way to analyze time-varying comparison data while generating clustered ranking results that facilitate more informed decision-making. Furthermore, we propose a combined penalty for group numbers and structure change points, allowing for the detection of long-term changes in underlying group configurations.

Several promising research directions remain open. First, extensions of the Bradley-Terry model that incorporate contextual information could be integrated into our framework to improve ranking accuracy. Second, while our current method focuses on pairwise comparisons, many practical scenarios involve comparisons among more than two candidates; extending the approach to handle such settings would broaden its applicability.

## Acknowledgments and Disclosure of Funding

We thank the area chairs and reviewers for their valuable comments and suggestions, which substantially improved this paper.

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

## A  Notations

We include Table 3 to summarize the key notations used throughout the paper.

Table 3: Notations

| Symbol | Description |
|--------|-------------|
| $[n]$ | Set of integers $\{1, \ldots, n\}$ |
| $\mathbf{I}$ | Identity matrix |
| $\boldsymbol{e}_n$ | $n \times 1$ vector with all elements equal to 1 |
| $\mathbf{0}$ | Zero vector or matrix |
| $\|\boldsymbol{v}\|_2$ | $\ell_2$-norm of vector $\boldsymbol{v}$ |
| $\|f\|_{2,T}$ | $L_2$ norm of function $f(\cdot)$ over the interval $[0, T]$ |
| $\mathcal{Y}$ | Set of pairwise comparison results |
| $y_{ij}(t)$ | Comparison result at time $t$ between items $i$ and $j$ |
| $\boldsymbol{\pi}^*(t)$ | Score vector of items at time $t$ in the Bradley-Terry model |
| $y_{ij}^*(t)$ | Winning probability between items $i$ and $j$ at time $t$ |
| $K(\cdot)$ | Kernel function |
| $\mathbf{P}(t)$ | Transformation probability matrix for the Markov chain |
| $B$ | Number of groups |
| $G$ | Group partition of items |
| $\delta_1$ | Minimum pairwise score difference between groups |
| $\delta_2$ | Maximum pairwise score difference within group |
| $\lambda$ | Tuning parameter for group recognition |
| $\mathbf{Q}$ | Constant matrix used for transformations |
| $\boldsymbol{\theta}(t)$ | Transformation of the score vector $\boldsymbol{\pi}(t)$ |
| $\mathbf{X}(t)$ | Matrix after transformation used in optimization objective for group recognition |
| $\boldsymbol{Y}(t)$ | Vector after transformation used in optimization objective for group recognition |
| $\boldsymbol{z}(t)$ | Latent group of items at time point $t$ |
| $\eta_i$ | Time points where the structure changes |
| $\xi_i$ | Candidate structure change points |
| $l(\boldsymbol{\pi}, t)$ | Negative log-likelihood function at time $t$ for score vector $\boldsymbol{\pi}$ |
| $\mathcal{P}$ | Partition of time interval |
| $\gamma_1, \gamma_2$ | Regularization parameters for the objective function |
| $\mathbf{A}^{\#}$ | Group inverse matrix |

## B  Supplementary to numerical results

### B.1  Experiment settings for ranking group recognition

For the experiments in Section 4.1, we consider two settings to evaluate our methods, as illustrated in Figure 4. We set $T = 1$, $B = 3$, and assign the number of items in each group as 3:3:4.

Define the first set as follows:

$$\boldsymbol{\pi}_i^*(t) - \text{pert}_i(t) = \begin{cases} \frac{1}{n}\big(2 + 0.3\sin(6\pi t)\big) & i \in G_1, \\ \frac{1}{n}\big(1 - 0.2\sin(6\pi t)\big) & i \in G_2, \\ \frac{1}{n}\big(0.25 - 0.075\sin(6\pi t)\big) & i \in G_3. \end{cases}$$

The $\text{pert}_i(t)$ is a perturbation term whose absolute value is less than $0.01/n$. For the second set, define it as:

$$\boldsymbol{\pi}_i^*(t) - \text{pert}_i(t) = \begin{cases} \frac{1}{n}\big(1.9 + 0.5\sin(3\pi t)\big) & i \in G_1, \\ \frac{1}{n}\big(0.1 + 0.6\arctan(\pi t)\big) & i \in G_2, \\ \frac{1}{n}\big(1 - 0.375\sin(3\pi t) - 0.45\arctan(\pi t)\big) & i \in G_3. \end{cases}$$

The point $\epsilon$ used for order estimation is $0.001$. The first setting represents a simple case, while the second setting is more complex with intersections of scores among different groups.

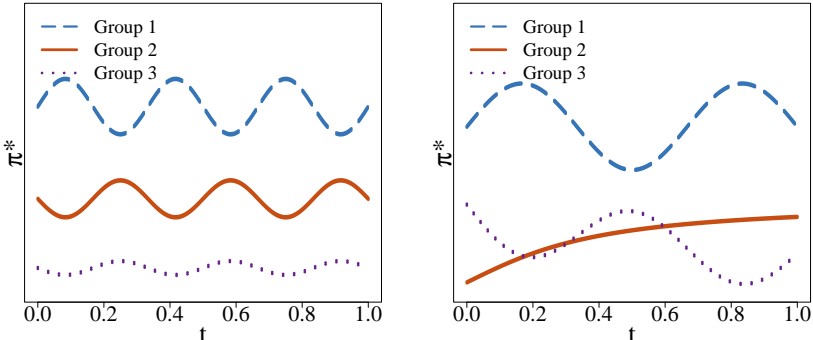

Figure 4: $\boldsymbol{\pi}^*(t)$ of each group in simulations.

Set $h = 0.05$, $m = 30$ and vary $n$ and $M$. We repeat each setting 500 times and use the extended BIC (EBIC) criterion [35, 22] to choose the tuning parameter $\lambda$. We note that cross-validation can be used here, but with the EBIC criteria, the computational cost is much lower. Specifically, we have

$$\text{EBIC}(\lambda) = nm \log\left(\frac{\text{RSS}(\lambda)}{nm} + c_0 \text{Var}(\boldsymbol{Y})\right) + \log(nm)\lceil df(\lambda) \rceil.$$

Here, $c_0 = 0.1$, $\lceil \cdot \rceil$ is the round down function. We let $\text{RSS}(\lambda) = \|\boldsymbol{Y} - \mathbf{X}\widehat{\boldsymbol{\theta}}(\lambda)\|_2^2$ and

$$df(\lambda) = \sum_{i=1}^{n-1} \mathbb{1}\{\|\widehat{\boldsymbol{\theta}}_i(\lambda)\|_2 > 0\} + \sum_{i=1}^{n-1} \frac{\|\widehat{\boldsymbol{\theta}}_i(\lambda)\|_2}{\|\widetilde{\boldsymbol{\theta}}_i\|_2}(m-1),$$

which is commonly used, for example, in [36, 37].

## B.2 Experiment settings for group changes recognition

For the problem of structural change detection in Section 4.2, we consider two settings: one with three stages and another with two stages. The true abilities are depicted in Figures 5 and 6, respectively. We set $h = 0.02$ and denote the observation times within each phase as $M$. The experiment is repeated 500 times. We employ the widely-used 10-fold cross-validation for the choice of tuning parameters $\gamma_1$ and $\gamma_2$.

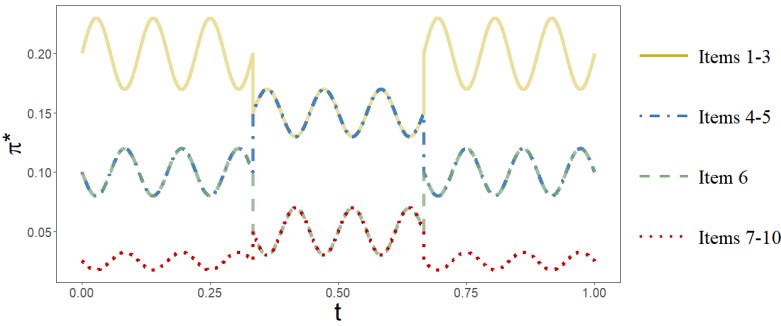

Figure 5: $\boldsymbol{\pi}^*(t)$ in the first setting.

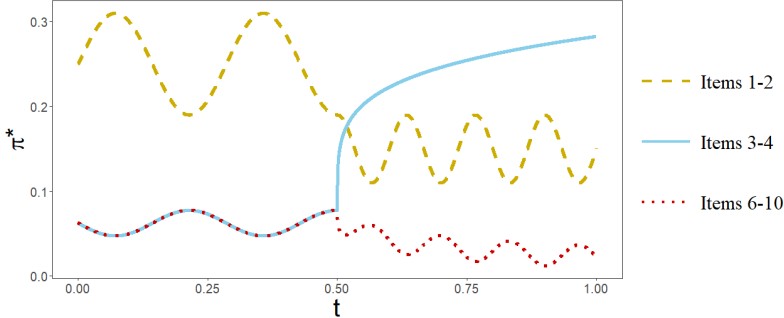

Figure 6: $\boldsymbol{\pi}^*(t)$ in the second setting.

Specifically, in setting 1, three phases are considered. The score functions in phases I and III remain the same, with a proportion of items being 3:3:4.

$$\boldsymbol{\pi}_i^*(t) = \begin{cases} 0.2 + 0.03\sin(18\pi t) & i = 1,2,3, \\ 0.1 - 0.02\sin(18\pi t) & i = 4,5,6, \\ 0.025 - 0.0075\sin(18\pi t) & i = 7,8,9,10. \end{cases}$$

For phase II, the item proportion is 1:1.

$$\boldsymbol{\pi}_i^*(t) = \begin{cases} 0.15 + 0.02\sin(18\pi t) & i = 1,\ldots,5, \\ 0.05 - 0.02\sin(18\pi t) & i = 6,\ldots,10. \end{cases}$$

In the second setting, during phase I, the item proportion is 1:4.

$$\boldsymbol{\pi}_i^*(t) = \begin{cases} 0.25 + 0.06\sin(7\pi t) & i = 1,2, \\ 0.0625 - 0.015\sin(7\pi t) & i = 3,\ldots,10. \end{cases}$$

For phase II, the item proportion is 1:1:3.

$$\boldsymbol{\pi}_i^*(t) = \begin{cases} 0.15 - 0.04\sin(15\pi t), & i = 1,2, \\ 0.065 + 0.25(t - \frac{1}{2})^{1/10}, & i = 3,4, \\ 0.1425 + 0.02\sin(15\pi t) - 0.0625(t - \frac{1}{2})^{1/10}, & i = 5,\ldots,10. \end{cases}$$

### B.3 Sensitivity of the hyperparameter choice

Hyperparameters play a crucial role in the performance of the proposed method, and their selection can be challenging. Ideally, computationally effective guidelines, such as those based on information criteria, would facilitate hyperparameter tuning. However, due to the novel optimization objective in our case, establishing such rules is non-trivial and warrants further investigation.

In this study, we employ cross-validation, which yields good empirical performance. To evaluate the sensitivity of the method to hyperparameter choices, we conduct additional experiments using parameter grids for the two experimental settings described in Section 4.2. Specifically, we repeat each combination of hyperparameters 50 times and calculate the average number of detected change points for each combination. The results are shown in Tables 4 and 5.

From the experimental results, we observe that the number of estimated change points varies with different values of $\gamma_1$ and $\gamma_2$. In particular, we find that $\gamma_1$ should be greater than $\gamma_2$ to ensure effective change point detection, consistent with our theoretical analysis in Theorem 3.9. Moreover, for a fixed $\gamma_1$, the estimated number of change points decreases as $\gamma_2$ increases. This aligns with our intuition, as larger values of $\gamma_2$ impose a higher penalty for each additional change point. Overall, while the tuning parameters affect the change point estimation, the estimated number of change points remains relatively stable even under substantial variations in the scales of $\gamma_1$ and $\gamma_2$.

### B.4 Supplementary to empirical analysis

For the empirical study, we set the bandwidth to match the season length, and other parameters remain consistent with those used in the simulations. Tuning parameters chosen through cross-validation

Table 4: Average change point number for different values of $\gamma_1$ and $\gamma_2$ for setting 1.

| $\gamma_1$ | 0.02 | 0.04 | 0.06 | 0.08 | 0.1 | 0.2 | 0.4 | 0.6 | 0.8 | 1 | 2 | 4 | 6 | 8 | 10 |
|---|---|---|---|---|---|---|---|---|---|---|---|---|---|---|---|
| $\gamma_2$=0.002 | 2.12 | 2.1 | 2.22 | 2.22 | 2.22 | 2.04 | 1.58 | 1.54 | 1.54 | 1.54 | 1.54 | 1.54 | 1.54 | 1.54 | 1.54 |
| $\gamma_2$=0.004 | 2.1 | 2.08 | 2.2 | 2.2 | 2.2 | 2.02 | 1.58 | 1.54 | 1.54 | 1.54 | 1.54 | 1.54 | 1.54 | 1.54 | 1.54 |
| $\gamma_2$=0.006 | 2.1 | 2.08 | 2.2 | 2.2 | 2.2 | 2.02 | 1.58 | 1.54 | 1.54 | 1.54 | 1.54 | 1.54 | 1.54 | 1.54 | 1.54 |
| $\gamma_2$=0.008 | 2.1 | 2.08 | 2.2 | 2.2 | 2.2 | 2.02 | 1.58 | 1.54 | 1.54 | 1.54 | 1.54 | 1.54 | 1.54 | 1.54 | 1.54 |
| $\gamma_2$=0.01 | 2.1 | 2.08 | 2.2 | 2.2 | 2.2 | 2.02 | 1.58 | 1.54 | 1.54 | 1.54 | 1.54 | 1.54 | 1.54 | 1.54 | 1.54 |
| $\gamma_2$=0.02 | 0 | 2.06 | 2.18 | 2.18 | 2.18 | 1.96 | 1.54 | 1.54 | 1.54 | 1.54 | 1.54 | 1.54 | 1.54 | 1.54 | 1.54 |
| $\gamma_2$=0.04 | 0 | 0 | 2.1 | 2.14 | 2.14 | 1.9 | 1.52 | 1.52 | 1.52 | 1.52 | 1.52 | 1.52 | 1.52 | 1.52 | 1.52 |
| $\gamma_2$=0.06 | 0 | 0 | 0 | 2.14 | 2.14 | 1.86 | 1.52 | 1.52 | 1.52 | 1.52 | 1.52 | 1.52 | 1.52 | 1.52 | 1.52 |
| $\gamma_2$=0.08 | 0 | 0 | 0 | 0 | 2.12 | 1.82 | 1.52 | 1.52 | 1.52 | 1.52 | 1.52 | 1.52 | 1.52 | 1.52 | 1.52 |
| $\gamma_2$=0.1 | 0 | 0 | 0 | 0 | 0 | 1.82 | 1.52 | 1.52 | 1.52 | 1.52 | 1.52 | 1.52 | 1.52 | 1.52 | 1.52 |
| $\gamma_2$=0.2 | 0 | 0 | 0 | 0 | 0 | 0 | 1.52 | 1.52 | 1.52 | 1.52 | 1.52 | 1.52 | 1.52 | 1.52 | 1.52 |
| $\gamma_2$=0.4 | 0 | 0 | 0 | 0 | 0 | 0 | 0 | 1.52 | 1.52 | 1.52 | 1.52 | 1.52 | 1.52 | 1.52 | 1.52 |
| $\gamma_2$=0.6 | 0 | 0 | 0 | 0 | 0 | 0 | 0 | 0 | 1.52 | 1.52 | 1.52 | 1.52 | 1.52 | 1.52 | 1.52 |
| $\gamma_2$=0.8 | 0 | 0 | 0 | 0 | 0 | 0 | 0 | 0 | 0 | 1.52 | 1.52 | 1.52 | 1.52 | 1.52 | 1.52 |
| $\gamma_2$=1 | 0 | 0 | 0 | 0 | 0 | 0 | 0 | 0 | 0 | 0 | 1.52 | 1.52 | 1.52 | 1.52 | 1.52 |

Table 5: Average change point number for different values of $\gamma_1$ and $\gamma_2$ for setting 2.

| $\gamma_1$ | 0.02 | 0.04 | 0.06 | 0.08 | 0.1 | 0.2 | 0.4 | 0.6 | 0.8 | 1 | 2 | 4 | 6 | 8 | 10 |
|---|---|---|---|---|---|---|---|---|---|---|---|---|---|---|---|
| $\gamma_2$=0.002 | 1.92 | 1.86 | 1.94 | 1.92 | 1.92 | 1.9 | 1.9 | 1.9 | 1.86 | 1.78 | 1.76 | 1.76 | 1.76 | 1.76 | 1.76 |
| $\gamma_2$=0.004 | 1.82 | 1.84 | 1.92 | 1.9 | 1.9 | 1.88 | 1.88 | 1.88 | 1.84 | 1.76 | 1.74 | 1.74 | 1.74 | 1.74 | 1.74 |
| $\gamma_2$=0.006 | 1.8 | 1.84 | 1.92 | 1.9 | 1.9 | 1.88 | 1.88 | 1.86 | 1.84 | 1.76 | 1.74 | 1.74 | 1.74 | 1.74 | 1.74 |
| $\gamma_2$=0.008 | 1.68 | 1.78 | 1.88 | 1.86 | 1.86 | 1.84 | 1.84 | 1.86 | 1.84 | 1.76 | 1.74 | 1.74 | 1.74 | 1.74 | 1.74 |
| $\gamma_2$=0.01 | 1.68 | 1.78 | 1.88 | 1.86 | 1.86 | 1.84 | 1.84 | 1.86 | 1.84 | 1.76 | 1.74 | 1.74 | 1.74 | 1.74 | 1.74 |
| $\gamma_2$=0.02 | 0 | 1.76 | 1.88 | 1.86 | 1.86 | 1.84 | 1.82 | 1.84 | 1.84 | 1.76 | 1.74 | 1.74 | 1.74 | 1.74 | 1.74 |
| $\gamma_2$=0.04 | 0 | 0 | 1.84 | 1.82 | 1.82 | 1.8 | 1.78 | 1.8 | 1.8 | 1.72 | 1.7 | 1.7 | 1.7 | 1.7 | 1.7 |
| $\gamma_2$=0.06 | 0 | 0 | 0 | 1.82 | 1.8 | 1.78 | 1.74 | 1.8 | 1.8 | 1.72 | 1.7 | 1.7 | 1.7 | 1.7 | 1.7 |
| $\gamma_2$=0.08 | 0 | 0 | 0 | 0 | 1.8 | 1.78 | 1.74 | 1.8 | 1.8 | 1.72 | 1.7 | 1.7 | 1.7 | 1.7 | 1.7 |
| $\gamma_2$=0.1 | 0 | 0 | 0 | 0 | 0 | 1.78 | 1.74 | 1.78 | 1.78 | 1.7 | 1.68 | 1.68 | 1.68 | 1.68 | 1.68 |
| $\gamma_2$=0.2 | 0 | 0 | 0 | 0 | 0 | 0 | 1.68 | 1.78 | 1.78 | 1.7 | 1.68 | 1.68 | 1.68 | 1.68 | 1.68 |
| $\gamma_2$=0.4 | 0 | 0 | 0 | 0 | 0 | 0 | 0 | 1.76 | 1.78 | 1.7 | 1.68 | 1.68 | 1.68 | 1.68 | 1.68 |
| $\gamma_2$=0.6 | 0 | 0 | 0 | 0 | 0 | 0 | 0 | 0 | 1.74 | 1.7 | 1.68 | 1.68 | 1.68 | 1.68 | 1.68 |
| $\gamma_2$=0.8 | 0 | 0 | 0 | 0 | 0 | 0 | 0 | 0 | 0 | 1.66 | 1.68 | 1.68 | 1.68 | 1.68 | 1.68 |
| $\gamma_2$=1 | 0 | 0 | 0 | 0 | 0 | 0 | 0 | 0 | 0 | 0 | 1.68 | 1.68 | 1.68 | 1.68 | 1.68 |

are $\gamma_1 = 0.04$ and $\gamma_2 = 0.006$. For the learning results demonstrated in Figure 2, in the first period, the GSW and SAS teams occupy the first two groups due to their extremely high winning rates. In the second phase, the first group includes GSW and SAS, while the second group consists of items from the top groups in the previous stage, with some changes. For instance, MEM and DAL shift to weaker groups. From the second to the third stage, a major change in the leading teams is notable: ATL, CLE, SAS disappear from the top groups, and teams like DEN, MIL, PHI emerge in the top 2 groups.

It is important to note that employing a static method yields significantly different results. For instance, in the initial phase, while also identifying seven groups, the static method GRC categorizes CHI into a single group, amalgamates POR into the MEM group, and groups all items from BOS to CHA (in the order presented in Figure 2, excluding POR) as a unified entity. Moreover, in the third phase, the static method recognizes three groups. The first group remains unchanged, with the subsequent ten items (excluding NOP but including LAC) forming the second group, while all other teams constitute the third group.

# C Technical proofs

## C.1 Proof of Theorem 3.3

*Proof.* Let $\mu(\mathbf{A})$ represent the eigenvalue of matrix $\mathbf{A}$. Let $\mathbf{A}_\mathcal{S}$ be the submatrix of $\mathbf{A}$ consisting of columns that correspond to items in $\mathcal{S}$ for matrix $\mathbf{A}$, and $x_\mathcal{S}$ be the subvector of $x$ comprising components corresponding to $\mathcal{S}$ for vector $x$. Before presenting the main theorem, we present the following lemma on the properties of $\mathbf{X}$ and $\mathbf{P}$.

**Lemma C.1.** Let Assumptions 3.1 and 3.2 hold. If $Mh \to \infty$, $n \to \infty$ and $nMh^5 \to 0$, we have $\|\mathbf{I} - \mathbf{P}(t)\|_2 = O_p(1)$, $\|(\mathbf{Q}^{-1})_{\mathcal{S}^c}\|_2 \lesssim \sqrt{\frac{B}{1+\cos\frac{(n-B)\pi}{n-B+1}}}$, $\mu_{\min}((\mathbf{X}_\mathcal{S})^\top(t)\mathbf{X}_\mathcal{S}(t)) \gtrsim \frac{n}{B}$ and $\|(\mathbf{X}_\mathcal{S}^\top(t)\mathbf{X}_\mathcal{S}(t))^{-1}\mathbf{X}_\mathcal{S}^\top(t)\|_2 \lesssim \sqrt{\frac{B}{n}}$ with probability tending to 1.

We first show the consistency of $\widetilde{\boldsymbol{\theta}}$. From Theorem S1 of Lu et al. [19], we have $\|\widetilde{\boldsymbol{\pi}}(t) - \boldsymbol{\pi}^*(t)\|_\infty = O_p(\delta)$. Combining the definition of $\boldsymbol{\theta}$, $\|\widetilde{\boldsymbol{\theta}}(t) - \boldsymbol{\theta}^*(t)\|_\infty = O_p(\delta)$.

Notice that

$$\widehat{\boldsymbol{\theta}} = \arg\min_{\boldsymbol{\theta}} Q(\boldsymbol{\theta}) = \arg\min_{\boldsymbol{\theta}} \frac{1}{2}\|Q - \mathbf{X}\boldsymbol{\theta}\|_2^2 + \lambda \sum_{i=1}^{n-1} \|\widetilde{\boldsymbol{\theta}}_i\|_2^{-1}\|\boldsymbol{\theta}_i\|_2. \tag{C.1}$$

Following the proof of Theorem 2.1 in [38], $Q(\boldsymbol{\theta})$ is a strictly convex function. Lemma 4.1 in [38] points out that, (C.1) is equivalent to

$$-\mathbf{X}_j^\top(\mathbf{Y} - \mathbf{X}\widehat{\boldsymbol{\theta}}) + \lambda\|\widetilde{\boldsymbol{\theta}}_j\|_2^{-1}\frac{\widehat{\boldsymbol{\theta}}_j}{\|\widehat{\boldsymbol{\theta}}_j\|_2} = \mathbf{0}, \forall \widehat{\boldsymbol{\theta}}_j \neq \mathbf{0},$$

and

$$\|\mathbf{X}_j^\top(\mathbf{Y} - \mathbf{X}\widehat{\boldsymbol{\theta}})\|_2 \leq \lambda\|\widetilde{\boldsymbol{\theta}}_j\|_2^{-1}, \forall \widehat{\boldsymbol{\theta}}_j = \mathbf{0},$$

where $\mathbf{X}_j$ represents the columns of $\mathbf{X}$ corresponding to $\boldsymbol{\theta}_j$. Therefore, it is sufficient to prove $\exists \boldsymbol{\theta}_0, \forall j \in \mathcal{S}, \boldsymbol{\theta}_{0j} \neq \mathbf{0}$, and $\forall j \notin \mathcal{S}, \boldsymbol{\theta}_{0j} = \mathbf{0}$, such that

$$-\mathbf{X}_j^\top(\mathbf{Y} - \mathbf{X}_\mathcal{S}\boldsymbol{\theta}_{0\mathcal{S}}) + \lambda\|\widetilde{\boldsymbol{\theta}}_j\|_2^{-1}\frac{\boldsymbol{\theta}_{0j}}{\|\boldsymbol{\theta}_{0j}\|_2} = \mathbf{0}, \forall j \in \mathcal{S}, \tag{C.2}$$

and

$$\|\mathbf{X}_j^\top(\mathbf{Y} - \mathbf{X}_\mathcal{S}\boldsymbol{\theta}_{0\mathcal{S}})\|_2 < \lambda\|\widetilde{\boldsymbol{\theta}}_j\|_2^{-1}, \forall j \notin \mathcal{S}. \tag{C.3}$$

Using (C.2), we have

$$-\mathbf{X}_\mathcal{S}^\top(\mathbf{Y} - \mathbf{X}_\mathcal{S}\boldsymbol{\theta}_{0\mathcal{S}}) + \lambda\beta_{0\mathcal{S}} = \mathbf{0},$$

where $\beta_{0\mathcal{S}} = (\frac{\boldsymbol{\theta}_{0j}^\top}{\|\widetilde{\boldsymbol{\theta}}_j\|_2\|\boldsymbol{\theta}_{0j}\|_2})_{j\in\mathcal{S}}^\top$. Notice that $\mathbf{X}_\mathcal{S}^\top\mathbf{X}_\mathcal{S}$ is invertible. Hence,

$$\begin{aligned}
\boldsymbol{\theta}_{0\mathcal{S}} &= (\mathbf{X}_\mathcal{S}^\top\mathbf{X}_\mathcal{S})^{-1}\mathbf{X}_\mathcal{S}^\top\mathbf{Y} - \lambda(\mathbf{X}_\mathcal{S}^\top\mathbf{X}_\mathcal{S})^{-1}\beta_{0\mathcal{S}} \\
&= \boldsymbol{\theta}_\mathcal{S}^* + ((\mathbf{X}_\mathcal{S}^\top\mathbf{X}_\mathcal{S})^{-1}\mathbf{X}_\mathcal{S}^\top\mathbf{Y} - \boldsymbol{\theta}_\mathcal{S}^*) - \lambda(\mathbf{X}_\mathcal{S}^\top\mathbf{X}_\mathcal{S})^{-1}\beta_{0\mathcal{S}}.
\end{aligned} \tag{C.4}$$

As for the second term,

$$\begin{aligned}
\|(\mathbf{X}_\mathcal{S}^\top\mathbf{X}_\mathcal{S})^{-1}\mathbf{X}_\mathcal{S}^\top\mathbf{Y} - \boldsymbol{\theta}_\mathcal{S}^*\|_\infty &= \|(\mathbf{X}_\mathcal{S}^\top\mathbf{X}_\mathcal{S})^{-1}\mathbf{X}_\mathcal{S}^\top(\mathbf{Y} - \mathbf{X}_\mathcal{S}\boldsymbol{\theta}_\mathcal{S}^*)\|_\infty \\
&\leq \sup_{k\in[m]} \|(\mathbf{X}_\mathcal{S}^\top(t_k)\mathbf{X}_\mathcal{S}(t_k))^{-1}\mathbf{X}_\mathcal{S}^\top(t_k)(\mathbf{Y}(t_k) - \mathbf{X}_\mathcal{S}(t_k)\boldsymbol{\theta}_\mathcal{S}^*(t_k))\|_2 \\
&\leq \sup_{k\in[m]} \|(\mathbf{X}_\mathcal{S}^\top(t_k)\mathbf{X}_\mathcal{S}(t_k))^{-1}\mathbf{X}_\mathcal{S}^\top(t_k)\|_2\|\mathbf{Y}(t_k) - \mathbf{X}_\mathcal{S}(t_k)\boldsymbol{\theta}_\mathcal{S}^*(t_k)\|_2,
\end{aligned} \tag{C.5}$$

where
$$\|\mathbf{Y}(t) - \mathbf{X}_{\mathcal{S}}(t)\boldsymbol{\theta}_{\mathcal{S}}^*(t)\|_2 \leq \|\mathbf{Y}(t) - \mathbf{X}_{-1}(t)\boldsymbol{\theta}^*(t)\|_2 + \|\mathbf{X}_{\mathcal{S}^c}(t)\boldsymbol{\theta}_{\mathcal{S}^c}^*(t)\|_2$$
$$= \|(\mathbf{P}^{*\top}(t) - \mathbf{P}^\top(t))\boldsymbol{\pi}^*(t)\|_2 + \|\mathbf{X}_{\mathcal{S}^c}(t)\boldsymbol{\theta}_{\mathcal{S}^c}^*(t)\|_2. \tag{C.6}$$

The proof of Theorem 1 in [22] implies
$$\|(\mathbf{P}^{*\top}(t) - \mathbf{P}^\top(t))\boldsymbol{\pi}^*(t)\|_2 = O_p(\sqrt{\frac{1}{n^2 Mh}}).$$

For the second term in (C.6),
$$\|\mathbf{X}_{\mathcal{S}^c}(t)\boldsymbol{\theta}_{\mathcal{S}^c}^*(t)\|_2 \leq \|\mathbf{X}_{\mathcal{S}^c}(t)\|_2 \|\boldsymbol{\theta}_{\mathcal{S}^c}^*(t)\|_2 \leq \|\mathbf{P}^\top(t) - \mathbf{I}\|_2 \|(Q^{-1})_{\mathcal{S}^c}\|_2 \|\boldsymbol{\theta}_{\mathcal{S}^c}^*(t)\|_2. \tag{C.7}$$

From Lemma C.1, we have $(C.7) \lesssim \sqrt{\frac{B}{1+\cos\frac{(n-B)\pi}{n-B+1}}}(n-B)\delta_2$. If $\delta_2 = o_p(\sqrt{\frac{1+\cos\frac{(n-B)\pi}{n-B+1}}{B(n-B)}\frac{1}{n^2 Mh}})$,

then (C.7) is $o_p(\sqrt{\frac{1}{n^2 Mh}})$. The first term in (C.5) is $O_p(\sqrt{\frac{B}{n}})$ using Lemma C.1. Hence, $(C.5)$ is

$O_p(\sqrt{\frac{B}{n^3 Mh}})$.

For the third term in (C.4), we have
$$\|\lambda(\mathbf{X}_{\mathcal{S}}^\top \mathbf{X}_{\mathcal{S}})^{-1}\beta_{0\mathcal{S}}\|_\infty \leq \sup_{k\in[m]} \|\lambda(\mathbf{X}_{\mathcal{S}}^\top(t_k)\mathbf{X}_{\mathcal{S}}(t_k))^{-1}\beta_{0\mathcal{S}}(t_k)\|_\infty$$
$$\leq \sup_{k\in[m]} \|\lambda(\mathbf{X}_{\mathcal{S}}^\top(t_k)\mathbf{X}_{\mathcal{S}}(t_k))^{-1}\beta_{0\mathcal{S}}(t_k)\|_2 \leq \sup_{k\in[m]} \lambda\|(\mathbf{X}_{\mathcal{S}}^\top(t_k)\mathbf{X}_{\mathcal{S}}(t_k))^{-1}\|_2 \|\beta_{0\mathcal{S}}(t_k)\|_2$$
$$\leq \lambda\frac{B}{n}\frac{\sqrt{B}}{\min_{i\in\mathcal{S}}\|\widetilde{\boldsymbol{\theta}}_i\|_2}. \tag{C.8}$$

Note that
$$\sqrt{\frac{1}{m}}\min_{i\in\mathcal{S}}\|\widetilde{\boldsymbol{\theta}}_i\|_2 \geq \sqrt{\frac{1}{m}}\min_{i\in\mathcal{S}}\|\boldsymbol{\theta}_i^*\|_2 - \sqrt{\frac{1}{m}}\max_{i\in\mathcal{S}}\|\widetilde{\boldsymbol{\theta}}_i - \boldsymbol{\theta}_i^*\|_2$$
$$\gtrsim \min_{i\in\mathcal{S}}\|\boldsymbol{\theta}_i^*(t)\|_{2,T} + O(\frac{1}{m}) + O_p(\delta) \gtrsim \delta_1.$$

Hence, $(C.8) \lesssim \frac{\lambda\sqrt{B^3}}{n\sqrt{m}\delta_1}$.

From (C.4), $\forall j \in \mathcal{S}$,
$$\|\boldsymbol{\theta}_{0j}\|_2 \gtrsim \sqrt{m}\delta_1 - \sqrt{m}\sqrt{\frac{B}{n^3 Mh}} - \frac{\lambda\sqrt{B^3}}{n\delta_1}.$$

If $\sqrt{\frac{B}{n^3 Mh\delta_1^2}} = o(1)$ and $\frac{\lambda B^{3/2}}{nm^{1/2}\delta_1^2} = o(1)$, then with probability tending to 1, we have $\forall j \in \mathcal{S}, \boldsymbol{\theta}_{0j} \neq$

$\mathbf{0}$. Actually, we have proved that if $\frac{\lambda B\sqrt{nMh}}{\sqrt{m}\delta_1} = O(1)$, then $\|\boldsymbol{\theta}_{0\mathcal{S}} - \boldsymbol{\theta}_{\mathcal{S}}^*\|_\infty = O_p(\sqrt{\frac{B}{n^3 Mh}})$.

Then we prove (C.3). Assume $j \notin \mathcal{S}$.
$$\|\mathbf{X}_j^\top(\mathbf{Y} - \mathbf{X}_{\mathcal{S}}\boldsymbol{\theta}_{0\mathcal{S}})\|_2 \leq \|\mathbf{X}_j^\top(\mathbf{Y} - \mathbf{X}_{\mathcal{S}}\boldsymbol{\theta}_{\mathcal{S}}^*)\|_2 + \|\mathbf{X}_j^\top(\mathbf{X}_{\mathcal{S}}\boldsymbol{\theta}_{\mathcal{S}}^* - \mathbf{X}_{\mathcal{S}}\boldsymbol{\theta}_{0\mathcal{S}})\|_2$$
$$\lesssim \sup_t \sqrt{m}\|\mathbf{X}_{\mathcal{S}^C}^\top(t)(\mathbf{Y}(t) - \mathbf{X}_{\mathcal{S}}(t)\boldsymbol{\theta}_{\mathcal{S}}^*(t))\|_\infty$$
$$+ \sqrt{m}\|(\mathbf{X}_{\mathcal{S}^C}^\top(t))(\mathbf{X}_{\mathcal{S}}(t)\boldsymbol{\theta}_{\mathcal{S}}^*(t) - \mathbf{X}_{\mathcal{S}}(t)\boldsymbol{\theta}_{0\mathcal{S}}(t))\|_\infty. \tag{C.9}$$

Similar to the proof of Theorem 1 in [28], we can obtain
$$(C.9) \leq \sup_t \sqrt{m}\max_i \|(\mathbf{X}_{\mathcal{S}^C}^\top(t))_{i\cdot}\|_2 \|(\mathbf{Y}(t) - \mathbf{X}_{\mathcal{S}}(t)\boldsymbol{\theta}_{\mathcal{S}}^*(t))\|_2$$
$$+ \sqrt{m}\max_i \|(\mathbf{X}_{\mathcal{S}^C}^\top(t)\mathbf{X}_{\mathcal{S}}(t))_{i\cdot}\|_2 \|\boldsymbol{\theta}_{\mathcal{S}}^*(t) - \boldsymbol{\theta}_{0\mathcal{S}}(t)\|_2$$
$$\leq \sup_t \sqrt{mn}\max_{i,k} |(\mathbf{X}_{\mathcal{S}^C}^\top(t))_{ik}| \|(\mathbf{P}^{*\top}(t) - \mathbf{P}^\top(t))\boldsymbol{\pi}^*(t)\|_2$$
$$+ \sqrt{mB}\max_{i,k} |(\mathbf{X}_{\mathcal{S}^C}^\top(t)\mathbf{X}_{\mathcal{S}}(t))_{ik}| \|\boldsymbol{\theta}_{\mathcal{S}}^*(t) - \boldsymbol{\theta}_{0\mathcal{S}}(t)\|_2. \tag{C.10}$$

The second inequality is gotten using (C.6). From the proof of Theorem 1 in [28], we have $\max_{i,k}|(\mathbf{X}_{\mathcal{S}^C}^\top(t))_{ik}| = O(1)$ and $\max_{i,k}|(\mathbf{X}_{\mathcal{S}^C}^\top(t)\mathbf{X}_{\mathcal{S}}(t))_{ik}| = O(n)$. Therefore, $(C.10) = O_p(\sqrt{\frac{mB^3}{nMh}})$.

On the other hand, since $\sqrt{\frac{1}{m}}\|\widetilde{\boldsymbol{\theta}}_j\|_2 \leq \sqrt{\frac{1}{m}}\|\widetilde{\boldsymbol{\theta}}_j - \boldsymbol{\theta}_j^*\|_2 + \sqrt{\frac{1}{m}}\|\boldsymbol{\theta}_j^*\|_2 \lesssim \delta_2 + \delta$, we have $\min_{j\notin\mathcal{S}}\lambda\|\widetilde{\boldsymbol{\theta}}_j\|_2^{-1} \gtrsim \frac{\lambda}{m^{1/2}\widetilde{\delta}}$. Since $\lambda \gtrsim \sqrt{\frac{m^2B^3}{nMh}\widetilde{\delta}}$, the theorem is proved. $\square$

## C.2 Proof of Theorem 3.6

*Proof.* Let $\mathbf{P}_G$ take the following form:
$$\mathbf{P}_{Gij}(t) = \begin{cases} \frac{1}{\widehat{B}}\frac{\sum_{l_1\in G_i}\sum_{l_2\in G_j}\sum_{t_k\in T_{l_1l_2}} y_{l_1l_2}(t_k)K_h(t,t_k)}{\sum_{l_1\in G_i}\sum_{l_2\in G_j}\sum_{t_k\in T_{l_1l_2}}K_h(t,t_k)}, & \text{if } i\neq j; \\ 1 - \sum_{s\neq i}\mathbf{P}_{Gis}(t), & \text{if } i = j. \end{cases}$$
Set $\widetilde{\boldsymbol{\pi}}_G$ as the stationary distribution of $\mathbf{P}_G$. Define $\mathbf{Y}_G(t) = B\mathbf{P}_G$. Let $\mathbf{Y}_G^* = B\mathbf{P}_G^*$. Similar to the proof of Theorem 1 in [28], using the derivative of stationary distribution, we can obtain
$$\frac{\partial\boldsymbol{\pi}_G^{*\top}(t)}{\partial\mathbf{Y}_{Gij}^*(t)} = \boldsymbol{\pi}_G^{*\top}(t)\frac{\partial\mathbf{P}_G^{*\top}(t)}{\partial\mathbf{Y}_{Gij}^*(t)}A^{\#}(t), \tag{C.11}$$
where
$$(\frac{\partial\mathbf{P}_G^{*\top}(t)}{\partial\mathbf{Y}_{Gij}^*(t)})_{ij} = (\frac{\partial\mathbf{P}_G^{*\top}(t)}{\partial\mathbf{Y}_{Gij}^*(t)})_{jj} = -(\frac{\partial\mathbf{P}_G^{*\top}(t)}{\partial\mathbf{Y}_{Gij}^*(t)})_{ji} = -(\frac{\partial\mathbf{P}_G^{*\top}(t)}{\partial\mathbf{Y}_{Gij}^*(t)})_{ii} = \frac{1}{B},$$
and other elements of the derivative matrix are zero.

Then we consider the difference between $\mathbf{Y}_G(t)$ and $\mathbf{Y}_G^*(t)$. For $i\neq j$,
$$\mathbf{Y}_{Gij}(t) - \mathbf{Y}_{Gij}^*(t)$$
$$=(\frac{\sum_{l_1\in G_i}\sum_{l_2\in G_j}\sum_{t_k\in T_{l_1l_2}} y_{l_1l_2}(t_k)K_h(t,t_k)}{\sum_{l_1\in G_i}\sum_{l_2\in G_j}\sum_{t_k\in T_{l_1l_2}}K_h(t,t_k)} - \frac{\sum_{l_1\in G_i}\sum_{l_2\in G_j}\sum_{t_k\in T_{l_1l_2}} y_{l_1l_2}^*(t_k)K_h(t,t_k)}{\sum_{l_1\in G_i}\sum_{l_2\in G_j}\sum_{t_k\in T_{l_1l_2}}K_h(t,t_k)})$$
$$+ (\frac{\sum_{l_1\in G_i}\sum_{l_2\in G_j}\sum_{t_k\in T_{l_1l_2}} y_{l_1l_2}^*(t_k)K_h(t,t_k)}{\sum_{l_1\in G_i}\sum_{l_2\in G_j}\sum_{t_k\in T_{l_1l_2}}K_h(t,t_k)} - \frac{1}{n_in_j}\sum_{l\in G_i}\sum_{k\in G_j}\frac{\boldsymbol{\pi}_k^*(t)}{\boldsymbol{\pi}_l^*(t) + \boldsymbol{\pi}_k^*(t)})$$
$$+ (\frac{1}{n_in_j}\sum_{l\in G_i}\sum_{k\in G_j}\frac{\boldsymbol{\pi}_k^*(t)}{\boldsymbol{\pi}_l^*(t) + \boldsymbol{\pi}_k^*(t)} - \mathbf{Y}_{Gij}^*(t))$$
$$=:(\mathbf{Y}_{Gij}(t) - \mathbf{Y}_{wij}^*(t)) + (\mathbf{Y}_{wij}^*(t) - \widetilde{\mathbf{Y}}_{Gij}^*(t)) + (\widetilde{\mathbf{Y}}_{Gij}^*(t) - \mathbf{Y}_{Gij}^*(t)). \tag{C.12}$$
Using central limit theorem, we have
$$\sqrt{n^2Mh}(\mathbf{Y}_{Gij} - \mathbf{Y}_{wij}^*) \xrightarrow{\mathcal{D}} N(0, \frac{1}{r_ir_j}\frac{\boldsymbol{\pi}_{G_i}^*(t)\boldsymbol{\pi}_{G_j}^*(t)}{(\boldsymbol{\pi}_{G_i}^*(t) + \boldsymbol{\pi}_{G_j}^*(t))^2}\int K^2(v)dv). \tag{C.13}$$
Notice that when $nMh^5 \to 0$,
$$\sqrt{n^2Mh}(\mathbf{Y}_{wij}^*(t) - \widetilde{\mathbf{Y}}_{Gij}^*(t))$$
$$=\sqrt{n^2Mh}(\frac{\sum_{l_1\in G_i}\sum_{l_2\in G_j}\sum_{t_k\in T_{l_1l_2}} y_{l_1l_2}^*(t_k)K_h(t,t_k)}{\sum_{l_1\in G_i}\sum_{l_2\in G_j}\sum_{t_k\in T_{l_1l_2}}K_h(t,t_k)} - \frac{1}{n_in_j}\sum_{l\in G_i}\sum_{k\in G_j}\frac{\boldsymbol{\pi}_k^*(t)}{\boldsymbol{\pi}_l^*(t) + \boldsymbol{\pi}_k^*(t)})$$
$$\to\sqrt{n^2Mh}(\frac{h^2\sum_{l\in G_i}\sum_{k\in G_j}\ddot{y}_{lk}^*(t)\int v^2K(v)\,dv}{2n_in_j}) \to 0. \tag{C.14}$$
Besides,
$$\sqrt{n^2Mh}|\widetilde{\mathbf{Y}}_G^*(t) - \mathbf{Y}_G^*(t)|$$
$$=\sqrt{n^2Mh}|\frac{1}{n_in_j}\sum_{l\in G_i}\sum_{k\in G_j}\frac{\boldsymbol{\pi}_k^*(t)}{\boldsymbol{\pi}_l^*(t) + \boldsymbol{\pi}_k^*(t)} - \frac{\boldsymbol{\pi}_{Gj}^*(t)}{\boldsymbol{\pi}_{Gi}^*(t) + \boldsymbol{\pi}_{Gj}^*(t)}|$$
$$\lesssim\sqrt{n^4Mh}\delta_2 \to 0. \tag{C.15}$$

Combining (C.11), (C.12), (C.13), (C.14) and (C.15), we have,

$$\sqrt{n^2 Mh}(\widetilde{\boldsymbol{\pi}}_G(t) - \boldsymbol{\pi}_G^*(t)) \xrightarrow{\mathcal{D}} \boldsymbol{\Gamma}(t) N(0, \boldsymbol{\Lambda}(t)). \tag{C.16}$$

Set $T_n(G) = \widetilde{\boldsymbol{\pi}}_G(t) - \boldsymbol{\pi}_G^*(t)$ and $T_n(\widehat{G}) = \widehat{\boldsymbol{\pi}}_G(t) - \boldsymbol{\pi}_G^*(t)$. From Theorem 3.3, $P(\widehat{G} = G) \to 1$. Therefore, for every $A \subset \mathbb{R}^B$, from

$$P(T_n(\widehat{G}) \in A) = P(T_n(\widehat{G}) \in A | \widehat{G} = G) P(\widehat{G} = G) + P(T_n(\widehat{G}) \in A | \widehat{G} \neq G) P(\widehat{G} \neq G),$$

we can obtian

$$\lim_{n \to \infty} P(T_n(\widehat{G}) \in A) = \lim_{n \to \infty} P(T_n(\widehat{G}) \in A | \widehat{G} = G) = \lim_{n \to \infty} P(T_n(G) \in A). \tag{C.17}$$

From (C.16) and (C.17), we have

$$\sqrt{n^2 Mh}(\widehat{\boldsymbol{\pi}}_G(t) - \boldsymbol{\pi}_G^*(t)) \xrightarrow{\mathcal{D}} \boldsymbol{\Gamma}(t) N(0, \boldsymbol{\Gamma}(t) \boldsymbol{\Lambda}(t) \boldsymbol{\Gamma}(t)^\top).$$

$\square$

## C.3  Proof of Theorem 3.9

*Proof.* We first prove that $P(\{\widehat{s}_i\}_{i=1}^{\widehat{J}} \supset \{\eta_i\}_{i=1}^J) \to 1$. Suppose there are change points in the interval $(\widehat{s}_i, \widehat{s}_{i+1})$, and assume the first one be $\eta_j$. Let $\mathcal{J}_1 = (\widehat{s}_i, \eta_j)$ and $\mathcal{J}_2 = (\eta_j, \widehat{s}_{i+1})$.

$$\left( \sum_{i=1}^2 (L(\widehat{\beta}(\mathcal{J}_i), \mathcal{J}_i) + \gamma_1 |\widehat{G}(\mathcal{J}_i)| |\mathcal{J}_i|) + \gamma_2 \right) - \left( L(\widehat{\beta}(\mathcal{J}_1 \cup \mathcal{J}_2), \mathcal{J}_1 \cup \mathcal{J}_2) + \gamma_1 |\widehat{G}(\mathcal{J}_1 \cup \mathcal{J}_2)| |\mathcal{J}_1 \cup \mathcal{J}_2| \right)$$
$$= \gamma_1 \left( |\widehat{G}(\mathcal{J}_1)| |\mathcal{J}_1| + |\widehat{G}(\mathcal{J}_2)| |\mathcal{J}_2| - |\widehat{G}(\mathcal{J}_1 \cup \mathcal{J}_2)| |\mathcal{J}_1 \cup \mathcal{J}_2| \right) + \gamma_2 + O_p(\delta_3). \tag{C.18}$$

Notice that

$$|\widehat{G}(\mathcal{J}_1)| |\mathcal{J}_1| + |\widehat{G}(\mathcal{J}_2)| |\mathcal{J}_2| - |\widehat{G}(\mathcal{J}_1 \cup \mathcal{J}_2)| |\mathcal{J}_1 \cup \mathcal{J}_2|$$
$$\to |G^*(\mathcal{J}_1)| |\mathcal{J}_1| + |G^*(\mathcal{J}_2)| |\mathcal{J}_2| - |G^*(\mathcal{J}_1 \cup \mathcal{J}_2)| |\mathcal{J}_1 \cup \mathcal{J}_2|,$$

and with the fact that $\mathcal{J}_1 \subset (\eta_{j-1}, \eta_j)$ and the right endpoint of $\mathcal{J}_1$ is exactly $\eta_j$, we have

$$|G^*(\mathcal{J}_1 \cup \mathcal{J}_2)| - |G^*(\mathcal{J}_1)| \geq 1.$$

Hence, with probability tending to 1,

$$(C.18) \leq \gamma_2 - \gamma_1 |\mathcal{J}_1| \leq \gamma_2 - \gamma_1 < 0,$$

which is contradictory to the definition of $\widehat{\mathcal{P}}$. Therefore, there are no change points between the estimated ones. In another word, $P(\{\widehat{s}_i\}_{i=1}^{\widehat{J}} \supset \{\eta_i\}_{i=1}^J) \to 1$.

The above proof classifies that all change points are in the estimation set, and we then prove that all points in the set are change points. Suppose that $\widehat{s}_i \notin \{\eta_i\}_{i=1}^J$, then there exists $j$ such that $\widehat{s}_i \in (\eta_j, \eta_{j+1})$. Let $\mathcal{J}_1 = (\eta_j, \widehat{s}_i)$ and $\mathcal{J}_2 = (\widehat{s}_i, \eta_{i+1})$.

Then we have

$$\left( \sum_{i=1}^2 (L(\widehat{\beta}(\mathcal{J}_i), \mathcal{J}_i) + \gamma_1 |\widehat{G}(\mathcal{J}_i)| |\mathcal{J}_i|) + \gamma_2 \right) - \left( L(\widehat{\beta}(\mathcal{J}_1 \cup \mathcal{J}_2), \mathcal{J}_1 \cup \mathcal{J}_2) + \gamma_1 |\widehat{G}(\mathcal{J}_1 \cup \mathcal{J}_2)| |\mathcal{J}_1 \cup \mathcal{J}_2| \right)$$
$$= \gamma_1 \left( |\widehat{G}(\mathcal{J}_1)| |\mathcal{J}_1| + |\widehat{G}(\mathcal{J}_2)| |\mathcal{J}_2| - |\widehat{G}(\mathcal{J}_1 \cup \mathcal{J}_2)| |\mathcal{J}_1 \cup \mathcal{J}_2| \right) + \gamma_2 + O_p(\delta_3) \to \gamma_2 > 0 \tag{C.19}$$

holds with probability tending to 1. Note that (C.19) means that removing $\widehat{s}_i$ in $\widehat{\mathcal{P}}$ leads to a strictly smaller value of (2.5), which is contradictory to the definition of $\widehat{\mathcal{P}}$. $\square$

## C.4 Proof of Corollary 3.10

*Proof.* First notice that $\frac{\|\boldsymbol{\pi}^*(t) - \widehat{\boldsymbol{\pi}}^{rf}(t)\|_2}{\|\boldsymbol{\pi}^*(t)\|_2} = O_p(\sqrt{\frac{1}{nMh}})$, $\forall t \in [0, V]$ by noticing the consistency of group estimation by Theorem 3.3 and convergence rate results in [22]. Then we only need to prove that if $\frac{\|\boldsymbol{\pi}(t) - \widetilde{\boldsymbol{\pi}}(t)\|_2}{\|\boldsymbol{\pi}(t)\|_2} = O_p(\zeta)$, $\forall t \in [0, V]$, then $\frac{1}{|\mathcal{I}|}|L(\boldsymbol{\pi}, \mathcal{I}) - L(\widetilde{\boldsymbol{\pi}}, \mathcal{I})| = O_p(\zeta)$. In fact, we can obtain

$$
\frac{1}{|\mathcal{I}|}|L(\boldsymbol{\pi}, \mathcal{I}) - L(\widetilde{\boldsymbol{\pi}}, \mathcal{I})| = \frac{1}{|\mathcal{I}|}|\int_{t \in \mathcal{I}} l(\boldsymbol{\pi}(t)) - l(\widetilde{\boldsymbol{\pi}}(t)) \, dt|
$$

$$
\leq \frac{1}{|\mathcal{I}|} \int_{t \in \mathcal{I}} \frac{2}{n(n-1)} \sum_{(i,j):i \neq j} \bar{y}_{ij}(t)| \log(\frac{\boldsymbol{\pi}_j(t)}{\boldsymbol{\pi}_i(t) + \boldsymbol{\pi}_j(t)}) - \log(\frac{\widetilde{\boldsymbol{\pi}}_j(t)}{\widetilde{\boldsymbol{\pi}}_i(t) + \widetilde{\boldsymbol{\pi}}_j(t)})| \, dt, \quad \text{(C.20)}
$$

where

$$
|\log(\frac{\boldsymbol{\pi}_j(t)}{\boldsymbol{\pi}_i(t) + \boldsymbol{\pi}_j(t)}) - \log(\frac{\widetilde{\boldsymbol{\pi}}_j(t)}{\widetilde{\boldsymbol{\pi}}_i(t) + \widetilde{\boldsymbol{\pi}}_j(t)})| = \log(1 + \frac{\widetilde{\boldsymbol{\pi}}_i(t)/\widetilde{\boldsymbol{\pi}}_j(t) - \boldsymbol{\pi}_i(t)/\boldsymbol{\pi}_j(t)}{1 + \boldsymbol{\pi}_i(t)/\boldsymbol{\pi}_j(t)})
$$

is $O_p(\zeta)$ using Taylor expansion. Then (C.20) is $O_p(\zeta)$. $\qquad \square$

## C.5 Proof of Lemma C.1

*Proof.* We first prove $\|\mathbf{I} - \mathbf{P}^*(t)\|_2 = O(1)$. We omit $t$ in this part for simplicity. Let $\boldsymbol{\pi}_0$ be the stationary distribution of $\mathbf{P}^*$, i.e., $\boldsymbol{\pi}_0^\top \mathbf{P}^* = \boldsymbol{\pi}_0$. Let $\Pi = diag(\boldsymbol{\pi}_0)$. Then under Assumption 3.1,

$$
\sqrt{\frac{\min_i \boldsymbol{\pi}_{0i}}{\max_i \boldsymbol{\pi}_{0i}}} \frac{\|\Pi^{1/2}(\mathbf{I} - \mathbf{P}^*)x\|_2}{\|\Pi^{1/2}x\|_2} \leq \frac{\|(\mathbf{I} - \mathbf{P}^*)x\|_2}{\|x\|_2} \leq \sqrt{\frac{\max_i \boldsymbol{\pi}_{0i}}{\min_i \boldsymbol{\pi}_{0i}}} \frac{\|\Pi^{1/2}(\mathbf{I} - \mathbf{P}^*)x\|_2}{\|\Pi^{1/2}x\|_2}.
$$

Hence, we have

$$
\|\mathbf{I} - \mathbf{P}^*\|_2 \asymp \max_{x \neq \mathbf{0}} \frac{\|\Pi^{1/2}(\mathbf{I} - \mathbf{P}^*)x\|_2}{\|\Pi^{1/2}x\|_2} \asymp \max_{x \neq \mathbf{0}} \frac{\|\Pi^{1/2}(\mathbf{I} - \mathbf{P}^*)\Pi^{-1/2}\Pi^{1/2}x\|_2}{\|\Pi^{1/2}x\|_2},
$$

which is the maximum singular value of $\Pi^{1/2}(\mathbf{I} - \mathbf{P}^*)\Pi^{-1/2}$. From 2.4.3 of [39], it is a symmetric matrix and its spectral norm equals $1 - \mu_{\min}(\mathbf{P}^*)$, which is $O(1)$. Further,

$$
\|\mathbf{I} - \mathbf{P}\|_2 \leq \|\mathbf{I} - \mathbf{P}^*\|_2 + \|\mathbf{P} - \mathbf{P}^*\|_2 = O_p(1)
$$

using Lemma 5 of [22].

Let $W$ be the inversion of $(\mathbf{I}^{-1})_{\mathcal{S}^c}^\top(\mathbf{Q}^{-1})_{\mathcal{S}^c}$. Similar to Lemma 3 in [28], $W$ is a $(n - B) \times (n - B)$ symmetric tridiagonal matrix, and for a vector $x$,

$$
x^\top W x \gtrsim \frac{1}{B}x_1^2 + \sum_{i=2}^{n-B} \frac{1}{B}(x_i - x_{i-1})^2 + \frac{1}{B}x_{n-B}^2,
$$

which is corresponding to a diagonal-constant matrix with the minimum eigenvalue being $\frac{2}{B}(1 + \cos \frac{(n-B)\pi}{n-B+1})$. Hence, $\|(\mathbf{Q}^{-1})_{\mathcal{S}^c}\|_2 \lesssim \sqrt{\frac{B}{1 + \cos \frac{(n-B)\pi}{n-B+1}}}$.

Since $\mathbf{X}^\top \mathbf{X}$ is a matrix with diagonal elements $\mathbf{X}_{-1}(t)^\top \mathbf{X}_{-1}(t)$, it is sufficient to prove the same for $\mathbf{X}_{-1}(t)$. Then we conclude the results using Lemma 4 in [28]. $\qquad \square$

# D Additional discussions

## D.1 Independence assumption of pairwise comparisons

We assume that pairwise comparisons, the elements of $\mathcal{Y}$, are independent in Section 2.1. The independence assumption among pairwise comparisons is a standard and widely adopted simplification in the BT model. In many real-world applications, such as sports tournaments and recommender systems, comparisons are typically collected independently across individuals or time, which renders the assumption approximately valid in practice.

For example, Masarotto and Varin [10] analyze outcomes from the National Football League and American College Hockey under the independence assumption. Similarly, Maystre et al. [40] apply a time-varying BT model to datasets from the NBA and the Association of Tennis Professionals (ATP), also assuming independence between matches. In the context of recommender systems, Liu et al. [8] adopt the BT framework for product ranking based on independent pairwise comparisons. All of these analyses demonstrate strong performance in practical settings.

Furthermore, recent studies in reinforcement learning from human feedback (RLHF) continue to employ the BT model with the independence assumption, which has proven effective. For instance, Xiong et al. [41], Zhong et al. [42], and Zhu et al. [43] develop RLHF algorithms using pairwise comparisons assumed to be independent, and report successful outcomes across various tasks.

## D.2 Smoothness condition of Assumption 3.1

We clarify that the smoothness condition in Assumption 3.1 is standard in dynamic settings and can be relaxed. In Section 2.2, it suffices for the score functions to be Lipschitz continuous to ensure consistent group identification (Theorem 3.3). The stronger smoothness assumption is only needed to derive the asymptotic distribution of the estimators (Theorem 3.6). Importantly, the Lipschitz condition is widely adopted in theoretical analyses of dynamic ranking problems, such as Assumption 5.2 of Bong et al. [29] and Assumption 1 of Karlé and Tyagi [44].

Furthermore, in Section 2.3, the general theoretical results in Theorem 3.9 rely only on Assumptions 3.7 and 3.8, which do not require the ability trajectories to be smooth. The smoothness condition in Corollary 3.10 is imposed solely to facilitate the application of Theorem 3.3, but as noted above, this can be weakened to a Lipschitz condition. Alternatively, one may use a segmented estimation strategy over a gridded time interval to estimate $\widehat{G}(\mathcal{I})$, and then apply our proposed change detection framework without requiring any smoothness assumption.

Finally, we would like to emphasize that our method performs well even in the presence of abrupt changes. This is supported by simulation results in Section 4.2, where the underlying score trajectories (shown in Figures 5 and 6 in Section B.2) feature nonsmooth and abrupt changes, yet our method maintains strong performance.

## D.3 Optimality of Theorem 3.6

Theorem 3.6 shows that the estimation error satisfies $\|\widehat{\boldsymbol{\pi}}_G(t) - \boldsymbol{\pi}_G^*(t)\|_\infty = O_p((n^2Mh)^{-1/2})$, which matches the optimal convergence rate. Specifically, for the refitted estimator $\widehat{\pi}_i^{rf}$ defined in equation (2.3), we obtain the relative error rate $\|\widehat{\boldsymbol{\pi}}^{rf}(t) - \boldsymbol{\pi}^*(t)\|_\infty / \|\boldsymbol{\pi}^*(t)\|_\infty = O_p((n^2Mh)^{-1/2})$. We analyze the result based on the effective sample size. On average, there are $n/B$ items per group, each compared against roughly $(B-1)n$ others. Assuming the Epanechnikov kernel with bandwidth $h$, each pair contributes about $2Mh$ effective observations. Hence, the total number of comparisons used to estimate each $\widehat{\pi}_i^{rf}(t)$ is of order $n^2Mh$. Since we pool comparisons across all items within the same group to estimate each ability score, the estimation procedure leverages this aggregated information. This matches the optimal order $L^{-1/2}$ established in Chen et al. [39] and Karlé and Tyagi [44], where $L$ denotes the average number of comparisons per item.

