# OpenReview forum: "Preference-Based Dynamic Ranking Structure Recognition"
_NeurIPS.cc/2025/Conference — NeurIPS 2025 poster_

### Official Review · Reviewer_Qzoa · 2025-07-01

**Clarity:** 1
**Significance:** 2
**Originality:** 4
**Rating:** 4
**Confidence:** 2

**Summary:**

This paper introduces a method for recognizing dynamic ranking structures in preference-based data, where instead of fixed scores, pairwise comparisons evolve over time.The authors build on the Bradley-Terry model and develop a technique that can:
- Group items that behave similarly over a time interval,
- Track how those groups change over time, and
- Rank the groups and their members in a meaningful way.
They tackle the challenges of noisy, time-varying data by combining spectral methods with optimization tricks like adaptive group lasso.
Experiments on NBA data show that their method produces interpretable, accurate rankings and captures real-world structural shifts like team trades.

**Questions:**

NA

**Ethical Concerns:**

["NO or VERY MINOR ethics concerns only"]

**Final Justification:**

The authors paritially address my concern.

**Limitations:**

Yes

**Quality:**

2

**Strengths And Weaknesses:**

Strengths:
- The paper introduces a dynamic framework for ranking and grouping, which is a step beyond static models and is very practical in real-world utility.
- The proposed method combines spectral methods, adaptive group lasso, and dynamic programming in a coherent way, and apprears to be theoratically guaranteed for group recognition and change point detection.

Weaknesses:
- The paper is very hard to read with multiple theorems, remarks, assumptions. This creates barriers for reader from understanding the content. Apart from this, one of the main contribution is confusing in lacking explanation of the terms "consistency recognition of groups", "structure recognition consistency", "uncertainty of item ability estimators". In addition, the figures are hard to read with font size being too small. Overall, I find great space left to improve the readability of the paper.
- An essential weakness of the paper is its strong reliance on smoothness and homogeneity assumptions in the latent ability trajectories. The method assumes that items within a group have similar, smoothly varying abilities over time, which may not hold in many real-world scenarios, especially in volatile domains like finance, user preferences, or competitive gaming. This assumption is critical to both the group identification and change point detection. If it is violated, the model's validity and interpretability could break down.

---

> ### Author Rebuttal · Authors · 2025-07-31
>
> We would like to thank you for your detailed and constructive comments. In light of the comments, we have revised the paper, which we believe has significantly improved both its quality and presentation. Below, we respond to the comments and suggestions point-by-point.
>
> 1. *"The paper is very hard to read with multiple theorems, remarks, assumptions. This creates barriers for reader from understanding the content. Apart from this, one of the main contribution is confusing in lacking explanation of the terms 'consistency recognition of groups', 'structure recognition consistency', 'uncertainty of item ability estimators'. In addition, the figures are hard to read with font size being too small. Overall, I find great space left to improve the readability of the paper."*
>
>    **Response:**
>    Thank you for pointing out this. In the revised version, we will enhance the readability of the paper by adding clearer explanations, smoother transitions, and more intuitive interpretations of the theoretical results and key terms.
>
>    Specifically, at the beginning of Section 3.1, we will add the following paragraph to clarify the main goal:  "We will present theoretical guarantees for our estimator in this section. Specifically, we show that the probability of identifying the correct underlying group structure approaches one as the sample size increases. We refer to this property as consistency recognition of groups, which is a desirable feature that supports the reliability of the estimation method."
>
>
>    After Theorem 3.3, we will include a transition to help guide the reader: "We have now established the consistency recognition of groups. The following two remarks clarify the conditions of the theorem and highlight the distinct features of our theoretical analysis."
>
>    In Section 3.2, we will add the following clarification at the beginning: "In this section, we discuss the uncertainty quantification, that is, the asymptotic distribution of the item ability estimators. A well-characterized uncertainty quantification allows for statistical inference tasks such as hypothesis testing and helps assess the reliability of the estimator."
>
>    In addition, at the beginning of Section 3.3, we will include this explanation: "In this section, we focus on structure recognition consistency, which refers to the probability of correctly identifying the group structure change points approaching one as the sample size increases. This property is important as it ensures reliable detection of structural changes in the data. We first provide a general analysis for arbitrary grouping methods in Theorem 3.10, and then specialize the discussion to the dynamic ranking setting in Corollary 3.11."
>
>    Additionally, we will improve the figures by enlarging the font size and adjusting the layout to enhance readability.
>
> 2. *"An essential weakness of the paper is its strong reliance on smoothness and homogeneity assumptions in the latent ability trajectories. The method assumes that items within a group have similar, smoothly varying abilities over time, which may not hold in many real-world scenarios, especially in volatile domains like finance, user preferences, or competitive gaming. This assumption is critical to both the group identification and change point detection. If it is violated, the model's validity and interpretability could break down."*
>
>    **Response:**
>    Thank you for your insightful comments.
>
>    First, we would like to clarify that the smoothness condition in Assumption 3.1 is standard in dynamic settings and can be relaxed.
>
>    In the recognition of dynamic ranking groups section, it is sufficient for the score functions to be Lipschitz continuous to ensure consistent group identification (Theorem 3.3). The stronger smoothness assumption is only used to derive the asymptotic distribution of the estimators (Theorem 3.6). Importantly, the Lipschitz condition is widely adopted in theoretical analysis of dynamic ranking problems, such as Assumption 5.2 of  Bong et al. (2020), and Assumption 1 of Karlé and Tyagi (2023).
>
>    Furthermore, in the recognition of group changes section, the general theoretical results in Theorem 3.10 rely only on Assumptions 3.8 and 3.9, which do not require the ability trajectories to be smooth. The smoothness condition in Corollary 3.11 is imposed solely to facilitate the application of Theorem 3.3, but as noted above, this can be weakened to a Lipschitz condition. Alternatively, one may use a segmented estimation strategy over a gridded time interval to estimate $\widehat G(\mathcal{I})$, and then apply our proposed group change detection framework, without requiring any smoothness assumption. We fully agree that nonsmooth, even abrupt, changes may occur in real-world scenarios. In fact, this is one of the main motivations behind our development of the group structure change detection framework, which is designed to effectively handle such irregularities in the data.
>
>    Secondly, the homogeneity condition (lines 115–116) is relatively mild. Many prior works assume that parameters are exactly equal within each group, for example, Assumption 1 in Tian and Shi (2023), whereas we allow for within-group variability and focus instead on identifying major structural differences.
>
>    Finally, we would like to emphasize that our method performs well even in the presence of abrupt changes. This is supported by simulation results in Section 4.2, where the underlying score trajectories (shown in Figures 4 and 5 in Section A.2) feature non-smooth and abrupt changes, yet our method maintains strong performance.
>
>    **References**
>    Bong, H., Li, W., Shrotriya, S., & Rinaldo, A. (2020, June). Nonparametric estimation in the dynamic Bradley-Terry model. In *International Conference on Artificial Intelligence and Statistics* (pp. 3317-3326). PMLR.
>
>    Karlé, E., & Tyagi, H. (2023). Dynamic ranking with the BTL model: a nearest neighbor based rank centrality method. *Journal of Machine Learning Research*, 24(269), 1-57.
>
>    Tian, X. Y., & Shi, J. (2023). Grouped rank centrality: Ranking and grouping from pairwise comparisons simultaneously. *Stat*, 12(1), e626.
>
> Again, thank you very much for your time and valuable comments.

---

> > ### Comment · Reviewer_Qzoa · 2025-08-06
> >
> > Thank you for the rebuttal. I appreciate the effort in improving the readability of the manuscript. I have adjusted my confidence score.

---

> > > ### Author Response · Authors · 2025-08-06
> > > **Appreciation for Your Valuable Feedback**
> > >
> > > Thank you for your feedback. We truly appreciate your careful consideration of our responses.
> > >
> > > Best regards,
> > >
> > > Authors

---

### Official Review · Reviewer_kwzX · 2025-07-01

**Clarity:** 2
**Significance:** 3
**Originality:** 3
**Rating:** 5
**Confidence:** 3

**Summary:**

This paper considers the problem of ranking structure recognition for preference-based data under the Bradley-Terry model, where the ranking structure could undergo changes over time. It first studies ranking group recovery for a given interval by introducing an optimization objective that can be formulated as the group lasso problem. Then, it applies the proposed clustering method to recognition of group changes by considering a combinatorial optimization problem. An algorithm based on dynamic programming is developed. Under certain technical assumptions, recognition consistency and estimator uncertainty results are established. Computational experiments are provided to evaluate the proposed framework.

**Questions:**

See above.

**Ethical Concerns:**

["NO or VERY MINOR ethics concerns only"]

**Limitations:**

Yes

**Quality:**

3

**Strengths And Weaknesses:**

Strengths:
The work proposes a framework for analyzing ranking structure recognition where the underlying cluster structure may experience changes, which is interesting and has potential applications to various areas.
It designs an objective function that allows for recovering the unknown change point of structures, and develops a DP-based algorithm to solve the combinatorial problem. I found this part inspiring.
Weakness:
While Theorem 3.6 quantifies the uncertainty of item ability estimations, can you comment on the optimality of the result?
Nonasymptotic recovery guarantees have received considerable attention in the recent literature of ranking. Do you expect such guarantees to hold for your proposed method?
Some notation choices may cause confusion for readers. For instance, on line 85: “\|f(t)\|_{2,T}=..., where $f(t)$ is a function on $[0,T]$”. Typically, $f(t)$ is perceived as a value and it might be more appropriate to use $f$ inside the norm.
On line 129: would it be ““induced by \tilde{\pi}” instead?

---

> ### Author Rebuttal · Authors · 2025-07-31
>
> We would like to thank you for your detailed and constructive comments. In light of the comments, we have revised the paper, which we believe has significantly improved both its quality and presentation. Below, we respond to the comments and suggestions point-by-point.
>
> 1. *"While Theorem 3.6 quantifies the uncertainty of item ability estimations, can you comment on the optimality of the result?"*
>
>    **Response:**
>    Thank you for your question. Theorem 3.6 shows that the estimation error satisfies $||\widehat{\mathbf{\pi}}\_{G}(t)-\mathbf{\pi}^{\*}\_{G}(t)||_{\infty} = O\_p((n^2Mh)^{-1/2})$, which matches the optimal rate. Specifically, for the refitted estimator $\widehat{\pi}_i^{\text{rf}}$ defined in equation (2.3), we obtain the relative error rate $||\widehat{\mathbf{\pi}}^{\text{rf}}(t)-\mathbf{\pi}^{\*}(t)||\_{\infty}/||\mathbf{\pi}^{*}(t)||\_{\infty} = O\_p((n^2Mh)^{-1/2})$. We analyze the result based on the effective sample size. Note that since we pool comparisons across all items in the same group to estimate each ability score, the estimation leverages this aggregated information. On average, there are $n/B$ items per group, each compared against roughly $(B-1)\cdot n$ others. Assuming the Epanechnikov kernel with bandwidth $h$, each pair contributes about $2Mh$ effective observations. Hence, the total number of comparisons used to estimate each $\widehat{\mathbf{\pi}}^{\text{rf}}_i(t)$ is of order $n^2Mh$. Therefore, the estimation rate $(n^2Mh)^{-1/2}$ matches the optimal rate $L^{-1/2}$ established in Chen et al. (2019) and Karlé and Tyagi (2023), where $L$ denotes the average number of comparisons per item.
>
>    **References**
>
>    Chen, Y., J. Fan, C. Ma, and K. Wang (2019). Spectral method and regularized MLE are both optimal for top-K ranking. *Annals of Statistics* 47 (4), 2204–223.
>
>    Karlé, E., & Tyagi, H. (2023). Dynamic ranking with the BTL model: a nearest neighbor based rank centrality method. *Journal of Machine Learning Research* 24 (269), 1–57.
>
> 2. *"Nonasymptotic recovery guarantees have received considerable attention in the recent literature of ranking. Do you expect such guarantees to hold for your proposed method?"*
>
>    **Response:**
>    Thank you for your insightful comment. Yes, we believe nonasymptotic recovery guarantees are attainable for our proposed methods. In Section 3.2, we introduce the group inverse of a Markov chain, which plays a key role in deriving nonasymptotic expansions for the spectral estimator. Building on Theorem S.1 in Lu et al. (2024), we can first expand the estimator expressed through the group inverse using properties of the Markov chain. Then, following Lemma S.1 in Lu et al. (2024), we approximate the group inverse using the relative ability scores. By substituting this approximation and carefully controlling the approximation error and remainder terms, we can obtain a nonasymptotic expansion of the estimator. While a rigorous derivation requires additional calculations, the direction is promising and deserves further study.
>
>    **References**
>
>    Lu, N., Shi, J., Tian, X.-Y., & Song, K. (2024). Tests on Dynamic Ranking. *Statistica Sinica*. DOI: 10.5705/ss.202024.0153
>
> 3. *"Some notation choices may cause confusion for readers. For instance, on line 85: '$||f(t)||\_{2,T}=...$, where $f(t)$ is a function on $[0,T]$'. Typically, $f(t)$ is perceived as a value and it might be more appropriate to use $f$ inside the norm."*
>
>    **Response:**
>    Thanks for pointing out this. We will revise it to $||f||\_{2,T}$.
>
> 4. *"On line 129: would it be 'induced by $\widetilde{\mathbf{\pi}}$' instead?"*
>
>    **Response:**
>    Thank you for your careful reading. We will correct it accordingly.
>
> Again, thank you very much for your time and valuable comments.

---

> > ### Comment · Reviewer_kwzX · 2025-08-05
> >
> > Thanks for addressing my questions.

---

> > > ### Author Response · Authors · 2025-08-06
> > > **Appreciation for Your Valuable Feedback**
> > >
> > > Thank you for your feedback. We truly appreciate your careful consideration of our responses.
> > >
> > > Best regards,
> > >
> > > Authors

---

### Official Review · Reviewer_RTDx · 2025-07-02

**Clarity:** 2
**Significance:** 3
**Originality:** 3
**Rating:** 4
**Confidence:** 2

**Summary:**

This paper proposes a novel framework for dynamic ranking structure recognition based on the Bradley-Terry model. It simultaneously identifies item groups with similar time-varying rankings and detects structural changes over time using a spectral estimation method combined with temporal penalties. The authors design an efficient optimization procedure using adaptive group lasso and dynamic programming, and provide theoretical guarantees for consistency and asymptotic distribution. Experiments on synthetic data and NBA game results demonstrate the method’s effectiveness and interpretability.

**Questions:**

1. Can the authors add one additional pargraph to briefly define the "group change",possibly with illustrative examples to enhance clarity?

2. Can the authors conduct additional experiments to evaluate the sensitivity of the method to hyperparameter choices?

**Ethical Concerns:**

["NO or VERY MINOR ethics concerns only"]

**Final Justification:**

I retain my positive score

**Limitations:**

The authors point out the limitation of current formulation only focuses on pairwise comparison, leaving the extension to multiple candidates open.

**Quality:**

3

**Strengths And Weaknesses:**

Strength:

This paper tries to tack the fundmental problem - how to track groups and rankings that change over time using preference data.  It introduces a novel, separable objective function that enables efficient optimization via dynamic programming. Theoretical results, including consistency and uncertainty quantification, are rigorously established. Empirical evaluations on synthetic and real-world datasets (e.g., NBA games) show that the proposed method outperforms existing baselines in both accuracy and interpretability.



Weakness:

The main weakness so far lies in the presentation. While the theoretical sections are mathematically sound, they are dense and lack intuitive explanation, making the paper difficult to follow. Including a complete notation table and illustrative diagrams could significantly enhance clarity.

Additionally, the authors mention using cross-validation to select hyperparameters—how sensitive is the method to these choices?

---

> ### Author Rebuttal · Authors · 2025-07-30
>
> We would like to thank you for your detailed and constructive comments. In light of the comments, we have revised the paper, which we believe has significantly improved both its quality and presentation. Below, we respond to the comments and suggestions point-by-point.
>
> 1. *"Can the authors add one additional paragraph to briefly define the 'group change', possibly with illustrative examples to enhance clarity?"*
>
>    **Response:**
>
>    Thank you for your insightful comment. To improve clarity, we have added the following paragraph in the revised version:
>
>    Many studies have addressed the grouping problem in ranked items, which involves assigning homogeneous items to the same group (e.g., in the context of journals, players, or sports teams). However, when considering the temporal aspect, we encounter situations where these groups evolve over time. For example, in basketball, player trades and coaching changes can significantly affect the team’s performance, potentially causing a team to rise to a higher ranking tier. In this context, the term 'group change' refers to these shifts in group membership over time, which are crucial for accurate analysis in the dynamic grouping problem.
>
> 2. *"Additionally, the authors mention using cross-validation to select hyperparameters—how sensitive is the method to these choices? Can the authors conduct additional experiments to evaluate the sensitivity of the method to hyperparameter choices?"*
>
>    **Response:**
>
>    Thank you for your question. Hyperparameters indeed play a crucial role in the performance of the method, and their selection can be challenging. Ideally, computationally effective guidelines, such as those based on information criteria like AIC, would help in choosing hyperparameters. However, due to the novel optimization objective in our case, establishing such rules is non-trivial and warrants further investigation.
>
>    In this study, we employ cross-validation, which yields good performance. To evaluate the sensitivity of the method to hyperparameter choices, we conduct additional experiments using parameter grids for the two experimental settings in Section 4.2. Specifically, we repeat each combination of hyperparameters 50 times and calculate the average number of change points for each combination. The results are shown in the tables below. (Due to response length limitations, we only report the results for Setting 1. The results for Setting 2 are similar, and we will include the full results in the revised version.)
>
>    **Estimated change point number for different values of $\gamma_1$ and $\gamma_2$ for setting 1:**
>
>    | $\gamma_1$          | 0.02 | 0.04 | 0.06 | 0.08 | 0.1  | 0.2  | 0.4  | 0.6  | 0.8  | 1    | 2    | 4    | 6    | 8    | 10   |
>    |---------------------|------|------|------|------|------|------|------|------|------|------|------|------|------|------|------|
>    | $\gamma_2$=0.002    | 2.12 | 2.1  | 2.22 | 2.22 | 2.22 | 2.04 | 1.58 | 1.54 | 1.54 | 1.54 | 1.54 | 1.54 | 1.54 | 1.54 | 1.54 |
>    | $\gamma_2$=0.004    | 2.1  | 2.08 | 2.2  | 2.2  | 2.2  | 2.02 | 1.58 | 1.54 | 1.54 | 1.54 | 1.54 | 1.54 | 1.54 | 1.54 | 1.54 |
>    | $\gamma_2$=0.006    | 2.1  | 2.08 | 2.2  | 2.2  | 2.2  | 2.02 | 1.58 | 1.54 | 1.54 | 1.54 | 1.54 | 1.54 | 1.54 | 1.54 | 1.54 |
>    | $\gamma_2$=0.008    | 2.1  | 2.08 | 2.2  | 2.2  | 2.2  | 2.02 | 1.58 | 1.54 | 1.54 | 1.54 | 1.54 | 1.54 | 1.54 | 1.54 | 1.54 |
>    | $\gamma_2$=0.01     | 2.1  | 2.08 | 2.2  | 2.2  | 2.2  | 2.02 | 1.58 | 1.54 | 1.54 | 1.54 | 1.54 | 1.54 | 1.54 | 1.54 | 1.54 |
>    | $\gamma_2$=0.02     | 0    | 2.06 | 2.18 | 2.18 | 2.18 | 1.96 | 1.54 | 1.54 | 1.54 | 1.54 | 1.54 | 1.54 | 1.54 | 1.54 | 1.54 |
>    | $\gamma_2$=0.04     | 0    | 0    | 2.1  | 2.14 | 2.14 | 1.9  | 1.52 | 1.52 | 1.52 | 1.52 | 1.52 | 1.52 | 1.52 | 1.52 | 1.52 |
>    | $\gamma_2$=0.06     | 0    | 0    | 0    | 2.14 | 2.14 | 1.86 | 1.52 | 1.52 | 1.52 | 1.52 | 1.52 | 1.52 | 1.52 | 1.52 | 1.52 |
>    | $\gamma_2$=0.08     | 0    | 0    | 0    | 0    | 2.12 | 1.82 | 1.52 | 1.52 | 1.52 | 1.52 | 1.52 | 1.52 | 1.52 | 1.52 | 1.52 |
>    | $\gamma_2$=0.1      | 0    | 0    | 0    | 0    | 0    | 1.82 | 1.52 | 1.52 | 1.52 | 1.52 | 1.52 | 1.52 | 1.52 | 1.52 | 1.52 |
>    | $\gamma_2$=0.2      | 0    | 0    | 0    | 0    | 0    | 0    | 1.52 | 1.52 | 1.52 | 1.52 | 1.52 | 1.52 | 1.52 | 1.52 | 1.52 |
>    | $\gamma_2$=0.4      | 0    | 0    | 0    | 0    | 0    | 0    | 0    | 1.52 | 1.52 | 1.52 | 1.52 | 1.52 | 1.52 | 1.52 | 1.52 |
>    | $\gamma_2$=0.6      | 0    | 0    | 0    | 0    | 0    | 0    | 0    | 0    | 1.52 | 1.52 | 1.52 | 1.52 | 1.52 | 1.52 | 1.52 |
>    | $\gamma_2$=0.8      | 0    | 0    | 0    | 0    | 0    | 0    | 0    | 0    | 0    | 1.52 | 1.52 | 1.52 | 1.52 | 1.52 | 1.52 |
>    | $\gamma_2$=1        | 0    | 0    | 0    | 0    | 0    | 0    | 0    | 0    | 0    | 0    | 1.52 | 1.52 | 1.52 | 1.52 | 1.52 |
>    ---
>
>    From the table above, we observe that the number of estimated change points varies with different values of $\gamma_1$ and $\gamma_2$. Specifically, we find that $\gamma_1$ should be greater than $\gamma_2$ to ensure effective change point detection, which is consistent with our theoretical analysis in Theorem 3.10. Moreover, for a fixed $\gamma_1$, the estimated number of change points decreases as $\gamma_2$ increases. This aligns with our intuition, as larger values of $\gamma_2$ imply a higher penalty for each additional change point. Overall, while there is some effect of the tuning parameters on the change point estimation, the estimated number of change points remains relatively stable, even with substantial variations in the scales of $\gamma_1$ and $\gamma_2$.
>
> 3. *"The main weakness so far lies in the presentation. While the theoretical sections are mathematically sound, they are dense and lack intuitive explanation, making the paper difficult to follow. Including a complete notation table and illustrative diagrams could significantly enhance clarity."*
>
>    **Response:**
>
>    Thank you for your comment. We appreciate the feedback. To enhance the readability and intuitiveness of the theoretical sections, we will make the following improvements in the revised version.
>
>    First, we will include the following table summarizing the key notations used throughout the paper.
>
>
>    **Notation table**
>    | **Symbol**        | **Description**                                            |
>    |-------------------|------------------------------------------------------------|
>    | $[n]$             | Set of integers $\{1, \ldots, n\}$                        |
>    | $\mathbf{I}$      | Identity matrix                                            |
>    | $\mathbf{e}_n$    | $n \times 1$ vector with all elements equal to 1           |
>    | $\mathbf{0}$      | Zero vector or matrix                                      |
>    | $\|\mathbf{v}\|_2$| $\ell_2$-norm of vector $\mathbf{v}$                       |
>    | $\|f\|_{2,T}$    | $L_2$ norm of function $f(\cdot)$ over the interval $[0,T]$|
>    | $\mathcal{Y}$     | Set of pairwise comparison results                         |
>    | $y_{ij}(t)$       | Comparison result at time $t$ between items $i$ and $j$    |
>    | $\mathbf{\pi}^*(t)$ | Score vector of items at time $t$ in the Bradley-Terry model |
>    | $y_{ij}^*(t)$     | Winning probability between items $i$ and $j$ at time $t$  |
>    | $K(\cdot)$        | Kernel function                                            |
>    | $\mathbf{P}(t)$   | Transformation probability matrix for the Markov chain     |
>    | $B$               | Number of groups                                           |
>    | $G$               | Group partition of items                                   |
>    | $\delta_1$        | Minimum pairwise score difference between groups           |
>    | $\delta_2$        | Maximum pairwise score difference within group             |
>    | $\lambda$         | Tuning parameter for group recognition                     |
>    | $\mathbf{Q}$      | Constant matrix used for transformations                   |
>    | $\mathbf{\theta}(t)$ | Transformation of the score vector $\mathbf{\pi}(t)$      |
>    | $\mathbf{X}(t)$   | Matrix after transformation used in optimization objective for group recognition |
>    | $\mathbf{Y}(t)$   | Vector after transformation used in optimization objective for group recognition |
>    | $\mathbf{z}(t)$   | Latent group of items at time point $t$                    |
>    | $\eta_i$          | Time points where the structure changes                    |
>    | $\xi_i$           | Candidate structure change points                          |
>    | $l(\mathbf{\pi}, t)$ | Negative log-likelihood function at time $t$ for score vector $\mathbf{\pi}$ |
>    | $\mathcal{P}$     | Partition of time interval                                  |
>    | $\gamma_1, \gamma_2$ | Regularization parameters for the objective function     |
>
>    In addition, we will add a diagram to illustrate the entire workflow of our methodology in the revised version. Although we cannot include the diagram here due to the rebuttal rules for this year, we describe it in detail as follows. The diagram will first depict the data formality, which will be represented by a comparison graph. This graph will show the preference results obtained from the pairwise comparisons. Next, we will illustrate the grouping method by using score curves, which show the score changes of each group and their relative ranking positions. Lastly, the structure detection procedure will be illustrated using a dynamic programming solution plot, structured as a tree to reflect the algorithm's design. In this plot, the nodes represent subproblems, the branches indicate recursive calls, and the leaves correspond to the base cases of our grouping method.
>
>    Besides, we will add more transitions and intuitive interpretations of the theoretical results and key terms to enhance the readability in the revised version.
>
> Again, thank you very much for your time and valuable comments.

---

> > ### Comment · Reviewer_RTDx · 2025-08-01
> >
> > I thank the authors for the detailed, point-to-point rebuttal. All my concerns have been solved. I retain my positive score.

---

> > > ### Author Response · Authors · 2025-08-06
> > > **Appreciation for Your Valuable Feedback**
> > >
> > > Thank you for your feedback. We truly appreciate your careful consideration of our responses.
> > >
> > > Best regards,
> > >
> > > Authors

---

### Official Review · Reviewer_Ha6P · 2025-07-07

**Clarity:** 3
**Significance:** 3
**Originality:** 4
**Rating:** 5
**Confidence:** 3

**Summary:**

The authors describe a statistical method to estimate a dynamic ranking structure in preference data. The Bradley-Terry model is used for inferring latent preference scores. Items are grouped together based on similar temporal behavior in their preferences. Given a grouping structure, the ranking order of these groups is determined. One of the goals of this study addresses the question how significant changes in the preference structure can be detected.

The method improves over SOTA techniques by higher accuracy and enhanced interpretability.

The objective function for scoring seems to be novel and it is designed to be solved by dynamic programming which guarantees efficient solutions. The Ranking Structure Recognition of NBA Teams shows fairly rapid changes in group structure associated with trading windows or the end of a season (see fig 2).

**Questions:**

In this paper, the basic assumption of the Bradlex-Terry model requires independence of the pair-wise preferences $y_{ij}(t_k) \in \mathcal{Y}$. How well does this assumption hold for real world preference data? It would be helpful for the reader to discuss the applicability of the BT model for today's applications.

Why does the normalization of the refit estimator in eq (2.3) scale with the absolute group size $|\hat G_k|$ and not with the relative group size?

What are the "separable properties" in the objective function that admit dynamic programming as a solution strategy? There seems to be only one occurence of "separable" in the manuscript stating the claim; but no explanation is given
why "separable properties" enable dynamic programming.

**Ethical Concerns:**

["NO or VERY MINOR ethics concerns only"]

**Final Justification:**

The answers of the authors are mostly satisfactory.

The motivation of the regularization term in eq (2.1) by fused lasso is still somewhat opaque to me. Statistical inference on rankings operates on the symmetric group with restrictions generated by the ranking data structure. Operating in this space can induce global constraints with implicit dependencies which casts some doubts on analogies to inference problems in Euclidean space.

Despite this somewhat critical remarks, I find the paper interesting and will stick to my original grading.

**Limitations:**

yes

**Quality:**

4

**Strengths And Weaknesses:**

Strengths:

Observing dynamic preferences requires a complex dynamics on the symmetric group to estimate a partial ranking. It couples grouping of objects when they are indistinguishable and ranking the resulting different groups. The model seems to perform well in applications when compared to SOTA methods.

The authors statistically analyze the group estimates $\hat{G}$ and prove consistency (Th 3.3) and asymptotic normality (Th 3.6). I find this analysis convincing.

The authors claim priority on the ranking structure changes for the BT model. The proposed objective function can be efficiently solved by dynamic programming due to its "separable properties". The experimental results document the improvements over SOTA results for group change detection.


Weaknesses:

The paper seems to be a compressed version of a longer manuscript and it reads a bit overloaded. I had to search together the definitions to understand the essence of the objective function and some of the statistical arguments.

- Some of the symbols are not clear, e.g., where is $\tilde{S}_k$ defined.

- It is also unclear (to me) why the normalization of the refit estimator in eq (2.3) scales with the absolute group size $|\hat G_k|$ and not with the relative group size.

- Algorithm 1 does not introduce a symbol for the observed data. I assumed it to be $mathcal{Y}$. Please add it to specify the data in the algorithm.

- The objective function (2.1) could be described in more detail. The first part seems to measure deviations from stationarity, while the $\lambda$ scaled part regularizes the inference. Why are ratios of item score differences proposed as regularization term? The regularization term often refers to the corresponding noise process of the function to be determined by variation. Does there exist such an interpretation for the second term in (2.1)?

---

> ### Author Rebuttal · Authors · 2025-07-30
>
> We would like to thank you for your detailed and constructive comments. In light of the comments, we have revised the paper, which we believe has significantly improved both its quality and presentation. Below, we respond to the comments and suggestions point-by-point.
>
> 1. *"In this paper, the basic assumption of the Bradley-Terry model requires independence of the pair-wise preferences $y_{ij}(t_k)\in\mathcal Y$. How well does this assumption hold for real world preference data? It would be helpful for the reader to discuss the applicability of the BT model for today's applications."*
>
>    **Response:**
>    Thank you for the insightful comment. The assumption of independence among pairwise comparisons is a standard and widely adopted simplification in the Bradley-Terry (BT) model. In many real-world applications, such as sports tournaments and recommender systems, comparisons are typically collected independently across individuals or time, which renders the assumption approximately valid in practice.
>
>    For example, Masarotto and Varin (2012) analyze outcomes from the National Football League and American College Hockey under the independence assumption. Similarly, Maystre et al. (2019) apply a time-varying BT model to datasets from the National Basketball Association (NBA) and the Association of Tennis Professionals (ATP), assuming independence between matches. In the context of recommender systems, Liu et al. (2023) adopt the BT framework for product ranking based on independent pairwise comparisons. All of these analyses demonstrate strong performance in practical settings.
>
>    Furthermore, recent studies in reinforcement learning from human feedback (RLHF) continue to employ the BT model with the independence assumption, which has proven effective. For instance, Xiong et al. (2023), Zhong et al. (2024), and Zhu et al. (2023) all develop RLHF algorithms using pairwise comparisons assumed to be independent, and report successful outcomes across various tasks.
>
>    While the independence assumption is widely adopted and often reasonable, we agree that extending the BT model to account for potential dependencies in comparisons is a valuable and promising direction for future research.
>
>    **References**
>    Liu, Y., Fang, E. X., & Lu, J. (2023). Lagrangian inference for ranking problems. *Operations Research*, 71(1), 202–223.
>    Masarotto, G., & Varin, C. (2012). The ranking lasso and its application to sport tournaments. *The Annals of Applied Statistics*, 6(4), 1949–1970.
>    Maystre, L., Kristof, V., & Grossglauser, M. (2019). Pairwise comparisons with flexible time-dynamics. *Proceedings of the 25th ACM SIGKDD International Conference on Knowledge Discovery & Data Mining*, 1236–1246.
>    Xiong, W., et al. (2023). Iterative preference learning from human feedback: Bridging theory and practice for RLHF under KL-constraint. *arXiv:2312.11456*.
>    Zhong, H., et al. (2024). Provable multi-party reinforcement learning with diverse human feedback. *arXiv:2403.05006*.
>    Zhu, B., Jordan, M., & Jiao, J. (2023). Principled reinforcement learning with human feedback from pairwise or k-wise comparisons. *International Conference on Machine Learning*, 43037-43067. PMLR.
>
> 2. *"Why does the normalization of the refit estimator in eq (2.3) scale with the absolute group size $|\widehat G\_k|$ and not with the relative group size? It is also unclear (to me) why the normalization of the refit estimator in eq (2.3) scales with the absolute group size $|\widehat G\_k|$ and not with the relative group size."*
>
>    **Response:**
>    Thank you for your question. As stated in Section 2.1, we impose the constraint that the sum of item abilities is 1, i.e., $\sum_{i=1}^n \pi_i^* = 1$ to ensure a unique representation without loss of generality. Since only the ratio between scores matters in the Bradley-Terry model, this constraint provides identifiability. When recovering the original scores after refitting, we still require that the normalized scores satisfy $\sum_{i=1}^n \widehat{\pi}_i^{\mathrm{rf}} = 1$. At the same time, we need to preserve the score ratios between items from different groups, meaning that for $i \in \widehat{G}_l$ and $j \in \widehat{G}_k$, we have $\frac{\widehat{\pi}\_i^{\mathrm{rf}}}{\widehat{\pi}\_i^{\mathrm{rf}} + \widehat{\pi}\_j^{\mathrm{rf}}} =\frac{\widehat{\pi}\_{\widehat{G}\_l}}{\widehat{\pi}\_{\widehat{G}\_l} + \widehat{\pi}\_{\widehat{G}\_k}}$. To satisfy both conditions simultaneously, the normalization in equation (2.3) is scaled by the absolute group size $|\widehat{G}_k|$. We will add this clarification in the revised version.
>
> 3. *"What are the 'separable properties' in the objective function that admit dynamic programming as a solution strategy? There seems to be only one occurence of 'separable' in the manuscript stating the claim; but no explanation is given why 'separable properties' enable dynamic programming."*
>
>    **Response:**
>    Thank you for your question. The term "separable properties" means that the objective function can be decomposed into independent subproblems. In our case, the objective is separable because it has an additive form across time, making it separable over time. Moreover, the objective satisfies the optimal substructure property, meaning that the optimal solution to the entire problem can be constructed from optimal solutions to its subproblems. These two properties, separability and optimal substructure, allow the objective to be optimized recursively, forming the basis for an efficient dynamic programming solution. We will add the explanation in the revised manuscript.
>
> 4. *"Some of the symbols are not clear, e.g., where is $\widetilde{\mathcal S}_k$ defined."*
>
>    **Response:**
>    Thank you for pointing out this. In the revised version, we will clarify the definition more clearly as follows.
>
>    We use $\widetilde{\mathcal S}$ to denote the estimated partition points of different groups. Specifically, we let $\widetilde{\mathcal S}=\{0\}\cup\widehat{\mathcal S}\cup\{n\}$. Without loss of generality, we assume $\widetilde{\mathcal S}$ is arranged in ascending order (if not, we simply reorder $\widehat S$), and $\widetilde{\mathcal S}_i$ denotes the $i$-th element of $\widetilde{\mathcal S}$. We will also include a notation table in the revised version to improve clarity.
>
> 5. *"Algorithm 1 does not introduce a symbol for the observed data. I assumed it to be $\mathcal{Y}$. Please add it to specify the data in the algorithm."*
>
>    **Response:**
>    Thank you for pointing this out. We will add $\mathcal{Y}$ to specify the observed data in Algorithm 1.
>
> 6. *"The objective function (2.1) could be described in more detail. The first part seems to measure deviations from stationarity, while the $\lambda$ scaled part regularizes the inference. Why are ratios of item score differences proposed as regularization term? The regularization term often refers to the corresponding noise process of the function to be determined by variation. Does there exist such an interpretation for the second term in (2.1)?"*
>
>    **Response:**
>    Thank you for the question. In equation (2.1), we use the score difference term $||\pi\_{i}(t) - \pi\_{i+1}(t)||\_{2,T}$ in the regularization. Although the form is more complex, it is inspired by the fused lasso penalty introduced by Tibshirani et al. (2005), where the term $\sum_{i}|\beta_i-\beta_{i-1}|$ is used to encourage similarity between adjacent coefficients. Regularization is typically employed to reduce the effect of noise and recover the underlying signal. In our context, the fused-type penalty serves a similar role: for homogeneous items that are expected to perform similarly, the penalty helps to reduce variability caused by noise within the same group.
>
>    Moreover, we incorporate an adaptive weight of the form $||\widetilde \pi_{i}(t)-\widetilde \pi_{i+1}(t)||_{2,T}^{-1}$, which is inversely proportional to the estimated score differences. This implies that item pairs with larger estimated differences, which likely reflect meaningful variation, are penalized less. In contrast, smaller estimated differences, which may be due to noise, are penalized more heavily. This adaptive weighting strategy, similar to the adaptive lasso (Zou, 2006), has been shown to improve empirical performance and provide desirable theoretical properties.
>
>    **References**
>    Tibshirani, R., Saunders, M., Rosset, S., Zhu, J., \& Knight, K. (2005). Sparsity and smoothness via the fused lasso. *Journal of the Royal Statistical Society Series B: Statistical Methodology*, 67(1), 91-108.
>
>    Zou, H. (2006). *Journal of the American Statistical Association*, 101(476), 1418-1429.
>
> Again, thank you very much for your time and valuable comments.

---

> > ### Comment · Reviewer_Ha6P · 2025-08-05
> > **Discussion of separability and the fused lasso penalty (point 3, 6)**
> >
> > Top 3:
> > The authors justify the separability with additivity over time. It is clear that eq (2.1) does not mix different times and this fact might be exploited for Dynamic Programming.
> >
> > What is not clear from the answer is the total computational effort which arises from the combinatorial tasks of solving the  subproblems at discrete time points $t_1, ... t_m$. The coupling between the $\pi_i$ might dominate the computation and then optimization over time is a side issue w.r.t. finding the optimal $\pi$ at a specific time point.
> >
> > Top 6:
> > The authors refer to fused lasso penalty of Tibshirani et al. (2005) as an inspiration for the regularization term in eq. (2.1).
> > However, there exists a significant difference between Tibshirani's original proposal and the ranking application in this paper.
> > Tibshirani's $\beta$s are elements of a Euclidean space and the $\pi$s are ranking probabilities of rankings that are elements of the symmetric group $\mathcal{S}_n$. $\mathcal{S}_n$ imposes global constraints that are not present in the Euclidean case. I find this issue is not adequately addressed in the manuscript. My question regarding  the underlying noise process is directed towards this issue since independent noise processes are limited by the degrees of freedom in $\mathcal{S}_n$ and that should be reflected in the regularization term.
> >
> > The other answers are satisfactory.

---

> ### Author Response · Authors · 2025-08-06
> **Discussion of separability and the fused lasso penalty (point 3, 6)**
>
> 1. *Discussion on point 3:*
>
>    **Response:**
>    Thank you for pointing this out. Yes, the primary cost arises from solving the subproblems. Specifically, the computational cost of solving the objective function (2.5) can be viewed as consisting of two layers.
>
>    The outer layer, as you pointed out, leverages the additivity over time. Therefore, for this layer, we apply the dynamic programming algorithm detailed in Algorithm 1 on Page 5.
>
>    For the inner layer, the subproblem involves calculating $\pi$. As you noted, this requires solving the challenging optimization problem in (2.1) to obtain $\widehat\pi$, which we transform into the equivalent form (2.2). Equation (2.2) corresponds to a standard adaptive group lasso problem, for which efficient solvers are available. We implement this step using the R package sparsegl.
>
>
> 2. *Discussion on point 6:*
>
>    **Response:**
>    Thank you for your comment. Our motivation for introducing the regularization term in equation (2.1) is to promote similarity between items, specifically to identify homogeneous groups. We refer to Tibshirani et al. (2005) as inspiration because both our formulation and the fused lasso share the idea of penalizing item differences to encourage grouping. In our case, $\pi$ represents ranking scores, which are subject to the constraint $\sum_{i=1}^{n}\pi_{i}(t_{k})=1, k=1,2,\ldots,m$, as shown in equation (2.1). To make the optimization of (2.1) more tractable, we apply a technical transformation using the matrix $\mathbf{Q}$, which eliminates the constraint and yields the simplified form in equation (2.2).
>
>
> Again, thank you very much for your discussions.

---

### Author Response · Authors · 2025-08-09
**Final Comment from Authors**

We sincerely thank the reviewers for their positive feedback, constructive suggestions, and insightful discussions, which have helped us improve the paper significantly.

Reviewers praised our preference-based dynamic ranking structure recognition framework as *introducing a novel objective function* (RTDx), *inspiring and having potential applications to various areas* (kwzX), *improving over SOTA techniques with higher accuracy and enhanced interpretability* (Ha6P), and *a step beyond static models that is very practical in real-world utility* (Qzoa). The theoretical analysis was also described as *convincing* (Ha6P), *rigorously established*, and *mathematically sound* (RTDx).

To address the main concern regarding clarity, we made several targeted improvements:
1. Added an illustrative diagram to enhance conceptual understanding.
2. Provided more intuitive interpretations and smoother transitions of the theoretical results and key terms.
3. Included a notation table to aid quick reference.

We have also carefully addressed all other questions and concerns raised by the reviewers. We appreciate the reviewers' time and thoughtful comments and believe the paper is now clearer, stronger, and more impactful thanks to the valuable input.

---

### Decision · Program_Chairs · 2025-09-17

**Decision:**

Accept (poster)

**Comment:**

The paper propose a ranking method that can track changes in preferences over time.

All reviewers agree that the paper makes interesting and publishable contributions. The main concern is that it is hard to read. The authors have promised a list of modifications to improve readability, and both the reviewers and I feel satisfied that these changes will be helpful. I urge the authors to make their best effort to implement them.